



# Simulating heat and $CO_2$ fluxes in Beijing using SUEWS V2020b: Sensitivity to vegetation phenology and maximum conductance

Yingqi Zheng[1,2,3], Minttu Havu[3], Huizhi Liu[1,2], Xueling Cheng[1,2], Yifan Wen[5], Hei Shing Lee[3,4], Joyson Ahongshangbam[3], and Leena Järvi[3,4]

[1]State Key Laboratory of Atmospheric Boundary Layer Physics and Atmospheric Chemistry, Institute of Atmospheric Physics, Chinese Academy of Sciences, Beijing, 100029, China
[2]University of Chinese Academy of Sciences, Beijing, 100029, China
[3]Institute for Atmospheric and Earth System Research/Physics, Faculty of Science, University of Helsinki, Helsinki, 00560, Finland
[4]Helsinki Institute of Sustainability Science, University of Helsinki, Helsinki, 00560, Finland
[5]School of Environment, State Key Joint Laboratory of Environment Simulation and Pollution Control, Tsinghua University, Beijing, 100084, China

**Correspondence:** Huizhi Liu (huizhil@mail.iap.ac.cn)

**Abstract.** The Surface Urban Energy and Water Balance Scheme (SUEWS) has recently been introduced to include a bottom-up approach to modelling carbon dioxide ($CO_2$) emissions and sink in urban areas. In this study, SUEWS is evaluated against radiation flux observations and eddy covariance (EC) measured turbulent fluxes of sensible heat ($Q_H$), latent heat ($Q_E$), and $CO_2$ ($F_C$) at a densely built neighborhood in Beijing. The model sensitivity to maximum conductance ($g_{max}$) and leaf area

5  index (LAI) is examined. Site-specific $g_{max}$ is obtained from observations over local vegetation species, and LAI parameters by optimization with remotely sensed LAI obtained from a MODIS/Terra data product. For simulation of anthropogenic $CO_2$ components, local traffic and population data are collected. In model evaluation, the mismatch between the measurement source area and simulation domain is also considered.

Using the optimized $g_{max}$ and LAI, the modelling of heat fluxes is noticeably improved, showing higher correlation with

10  observations, lower bias, and more realistic seasonal dynamics of $Q_E$ and $Q_H$. In comparison to heat fluxes, the $F_C$ module shows lower sensitivity to the choice of $g_{max}$ and LAI. This can be explained by the low relative contribution of vegetation to net $F_C$ in the modelled area. SUEWS successfully reproduces the average diurnal cycle of $F_C$ and annual cumulative sums. Depending on the size of the simulation domain, the modelled annual accumulated $F_C$ ranges from 7.2 to 8.5 kg C $m^{-2}$ $yr^{-1}$, when compared to 7.5 kg C $m^{-2}$ $yr^{-1}$ observed by EC. Traffic is the dominant $CO_2$ source, contributing 63–73% to

15  the annual total $CO_2$ emissions, followed by human metabolism (14–18%), respiration released by vegetation and soil (6–11%) and building heating (6–9%). Vegetation photosynthesis offsets only 4–8% of the total $CO_2$ emissions. We highlight the importance of choosing optimal LAI parameters and $g_{max}$ when SUEWS is used to model surface fluxes. The $F_C$ module of SUEWS is a promising tool in quantifying urban $CO_2$ emissions at the local scale, and therefore assisting to mitigate urban $CO_2$ emissions.



## 1  Introduction

Currently, half of the global population resides in urban areas, and this percentage is projected to grow to 68% by the middle of the 21$^{st}$ century (United Nations Department of Economic and Social Affairs, 2019). The urban expansion reshapes the morphological, thermal, and dynamical properties of the land surface (Grimmond and Oke, 2006; Oke, 1995; Zhu et al., 2016), and intensive human activities contribute to a major source of greenhouse gas emissions (Marcotullio et al., 2013; Velasco and Roth, 2010). This has influenced urban climate from micro to regional scales (Johansson and Emmanuel, 2006; Sarangi et al., 2018; Tan et al., 2010). Climatic and environmental risks due to urbanization are frequently reported, such as heat waves, flooding and air pollution (Qian et al., 2022; Watts et al., 2015). In this context, there is a pressing need to better understand the effects of urbanization on land-atmosphere interaction, preferably in a quantitative way.

Urban land surface models (ULSMs) are widely used to simulate urban-atmosphere interactions, including the exchanges of energy, water and $CO_2$, and hydrological processes (Chen et al., 2011; Masson et al., 2013). The results from the First International Urban Land Surface Model Comparison Project suggest that the most important processes for surface energy balance in the urban environment were radiative and vegetation processes (e.g. surface fraction of vegetation, seasonal cycle of vegetation phenology) (Grimmond et al., 2010; Best and Grimmond, 2015; Nordbo et al., 2015). Long-term observations with low vegetation cover (<30%) were especially needed to evaluate heat flux simulation, as energy distribution was found sensitive in such environments (Best and Grimmond, 2016).

The Surface Urban Energy and Water balance Scheme (SUEWS) is one of the widely tested ULSMs (Järvi et al., 2011, 2014; Ward et al., 2016). SUEWS is designed to run with surface information (e.g., surface cover fractions) and a minimum amount of model forcing data. In recent years, Supy (SUEWS in Python) was developed to allow Python front-end implementation for broader and easier applications (Sun and Grimmond, 2019). SUEWS has demonstrated good performance against hydrological observations and surface flux observations in several cities in Europe, North America, and Asia (Järvi et al., 2011; Alexander et al., 2016; Ward et al., 2016; Ao et al., 2018; Havu et al., 2022a). In SUEWS, seasonal cycle of vegetation phenology is indicated by leaf area index (LAI). Previous studies made in two UK cities and Shanghai, China have reported that bias in modelled LAI leads to over- or underestimation in $Q_E$ or $Q_H$ (Ao et al., 2018; Ward et al., 2016). They highlight the importance of having an appropriate seasonal cycle of LAI in SUEWS. Omidvar et al. (2022) proposed a workflow to derive LAI-related parameters for SUEWS, but it was intended for fully vegetated areas located mainly on the outskirt of cities. Apart from LAI, the maximum conductance ($g_{max}$) is also critical in scaling the surface conductance ($g_s$), and therefore the available energy distribution (Ward et al., 2016). However, the impact of LAI-related parameters and $g_{max}$ on modelled turbulent fluxes has received insufficient attention in urban areas.

Recently, the module of local-scale $CO_2$ flux ($F_C$) was incorporated into SUEWS (Järvi et al., 2019). It was found to give reasonable annual sum, seasonal and diurnal cycles against observed $F_C$ in Helsinki, Finland, suggesting that the bottom-up $CO_2$ emission or sink estimate in SUEWS can be evaluated by observation-based evidence provided by top-down eddy covariance (EC) measurements. Furthermore, SUEWS shows the potential for broader use, such as quantifying the carbon sequestration potential of urban vegetation (Havu et al., 2022a), investigating the spatial variability of $CO_2$ emission, quanti-



fying the contribution of each emission component (Järvi et al., 2019), and therefore assisting urban $CO_2$ emission mitigation.
However, this module has not yet been evaluated in other cities than Helsinki.

Beijing provides a unique test-bed for SUEWS evaluation and application: a mega-city with population of over 21 million and an increasing urbanized area (MHURD, 2018). The older version of SUEWS (V2017b), has been evaluated and applied in Beijing by Kokkonen et al. (2019), showing good model performance against observed heat fluxes. However, good simulation of turbulent flux does not necessarily imply the sub-models within give accurate estimates, e.g., LAI and radiative components.
Correct presentation of these processes is necessary for more advanced applications such as prediction of surface exchanges of energy under e.g., future climate scenarios. Besides, the newly-developed $F_C$ module has not yet been evaluated in Beijing.

In this paper, we present a comprehensive evaluation of SUEWS V2020b in simulating surface fluxes of energy and $CO_2$ in Beijing. The main aims of this study are (1) to evaluate the model performance of SUEWS using different vegetation parameters (default and site-specific) against radiation and turbulent flux ($Q_E$, $Q_H$ and $F_C$) measurements, and (2) to partition $F_C$ into
different anthropogenic and biogenic components and quantify the contribution of each component. Meanwhile, the impact of the mismatch between the turbulent flux source area and the modelled area is also examined.

## 2 Model Description

SUEWS is an urban land surface model that simulates the surface energy and water balances, and $CO_2$ flux at a local (neighborhood) scale (Järvi et al., 2011; Ward et al., 2016; Järvi et al., 2019). In SUEWS, the modelling domain is separated into
seven interacting surface types (buildings, paved surfaces, grass, evergreen trees/shrub, deciduous trees/shrubs, bare soil, and water body), with a single soil layer below each type. SUEWS is designed to run with surface information (e.g., surface cover fractions) and a minimal amount of model forcing data including wind speed ($U$), relative humidity ($RH$), air temperature ($T_{air}$), air pressure ($p$), precipitation and incoming solar radiation ($K_{down}$). SUEWS has sub-models for LAI and net all-wave radiation, and users are allowed to modify the parameters of the sub-models based on information of the modelled domain. In
this study, we use SUEWS version V2020b (Havu et al., 2022b).

### 2.1 Leaf area index model

In SUEWS, leaf growth is accumulated when $T_{air}$ stays above limit value $T_{base,GDD,i}$ consecutively, denoted by growing-degree-day (GDD). Leaf growth is allowed until GDD reaches the upper boundary $GDD_{full,i}$ or LAI reaches its maximum ($LAI_{max,i}$). Similarly to leaf growth period, leaf off period is impacted by $T_{base,SDD,i}$, senescence-degree-day (SDD), and
$SDD_{full,i}$ or $LAI_{min,i}$. Leaf fall is controlled by $T_{air}$ or at high latitudes by day length (Järvi et al., 2014). Here, LAI for vegetation type $i$ at the day of year $d$ ($LAI_{d,i}$) is defined as:

$$LAI_{d,i} = \begin{cases} \min\left(LAI_{max,i}, LAI_{d-1,i}^{\omega_1,GDD,i} \cdot GDD_{d,i} \cdot \omega_{2,GDD,i} + LAI_{d-1,i}\right), \text{leaf-on}, T_{base,GDD,i} > T_{d-1} \\ \max\left(LAI_{min,i}, LAI_{d-1,i}^{\omega_1,SDD,i} \cdot SDD_{d,i} \cdot \omega_{2,SDD,i} + LAI_{d-1,i}\right), \text{leaf-off}, T_{d-1} < T_{base,SDD,i}, \end{cases} \tag{1}$$





where $LAI_{max,i}$ and $LAI_{min,i}$ for each vegetation type can be obtained from literature or determined from observations, $\omega_{1/2,GDD/SDD,i}$ represent the growing or senescence rates derived for each study site or use their default values (Järvi et al., 2011; Omidvar et al., 2022), and $T_{d-1}$ is the previous day air temperature mean.

## 2.2 Radiation fluxes

$K_{down}$ is a required variable in the meteorological forcing data, whereas other radiation components are estimated within SUEWS. Outgoing shortwave radiation ($K_{up}$) is calculated using a bulk albedo ($\alpha$) based on the area fraction for each surface type. Incoming longwave radiation ($L_{down}$) is calculated using $T_{air}$ and $RH$ to estimate the cloud cover and the clear-sky emissivity (Loridan et al., 2011), while outgoing longwave radiation ($L_{up}$) is estimated by surface emissivity, $\alpha$, $K_{down}$, $L_{up}$ and $T_{air}$ (Offerle et al., 2003).

## 2.3 Turbulent heat fluxes

Latent heat flux ($Q_E$, W m$^{-2}$) is calculated using the modified Penman-Monteith equation for each surface type:

$$Q_E = \frac{s(Q_N + Q_F - \Delta Q_s) + \rho c_p VPD/r_{av}}{s + \gamma(1 + r_s/r_{av})}, \tag{2}$$

where $Q_N$ (W m$^{-2}$) is the net all-wave radiation, $Q_F$ (W m$^{-2}$) the anthropogenic heat flux, $\Delta Q_S$ (W m$^{-2}$) the net storage heat flux, $\rho$ (kg m$^{-3}$) the air density, $c_p$ (J kg$^{-1}$ K$^{-1}$) the specific heat capacity of air at constant pressure, $VPD$ (Pa) the vapour pressure deficit, $s$ (Pa $^\circ$C$^{-1}$) the slope of the saturation vapour pressure curve, $\gamma$ (Pa $^\circ$C$^{-1}$) the psychrometric constant, $r_{av}$ (s mm$^{-1}$) the aerodynamic resistance for water vapour, and $r_s$ (s mm$^{-1}$) the surface resistance. $r_s$, or its inverse surface conductance $g_s$ (mm s$^{-1}$), has the form:

$$g_s = \frac{1}{r_s} = \sum_i (g_{max,i} \frac{LAI_i}{LAI_{max,i}} fr_i) G_1 g(K_{down}) g(\Delta q) g(T_{air}) g(\Delta \theta), \tag{3}$$

where $g_{max,i}$ is the maximum conductance of vegetation type $i$, $fr_i$ is the surface fraction of $i$, $G_1$ is a constant connecting stomatal conductance to canopy conductance, $g(K_{down})$, $g(\Delta q)$, $g(T_{air})$, and $g(\Delta \theta)$ are environmental response functions on $K_{down}$, specific humidity deficit ($\Delta q$), air temperature ($T_{air}$), and soil moisture deficit ($\Delta \theta$), respectively. The functions have forms (Ward et al., 2016):

$$g(K_{down}) = \frac{K_{down}/(G_2 + K_{down})}{K_{down,max}/(G_2 + K_{down,max})}, \tag{4}$$

$$g(\Delta q) = G_3 + (1 - G_3) G_4^{\Delta q}, \tag{5}$$

$$g(T_{air}) = \frac{(T_{air} - T_L)(T_H - T_{air})^{T_C}}{(G_5 - T_L)(T_H - G_5)^{T_C}}, \tag{6}$$





where

$$T_C = \frac{(T_H - G_5)}{(G_5 - T_L)}, \tag{7}$$

and

$$g(\Delta\theta) = \frac{1 - \exp(G_6(\Delta\theta - \Delta\theta_{WP}))}{1 - \exp(-G_6\Delta\theta_{WP})}. \tag{8}$$

Coefficients $G_2 - G_6$ determine the shape of the curves describing responses of stomatal conductance to each environmental variable. $K_{down,max}$ (W m$^{-2}$) is the maximum incoming solar radiation, $T_L$ and $T_H$ (°C) are the lower and upper limits for temperature at which photosynthesis and transpiration are off, and $\Delta\theta_{WP}$ (mm) is wilting point deficit. $K_{down}$ (W m$^{-2}$) is model input, $\Delta q$ (g kg$^{-1}$) is calculated from model input $RH$, $T_{air}$ (°C) is either model input or simulated at 2 m height, and $\Delta\theta$ (mm) is simulated within SUEWS (Järvi et al., 2017). $Q_H$ is determined as the residual from the surface energy balance equation:

$$Q_H = Q_N + Q_F - \Delta Q_s - Q_E. \tag{9}$$

## 2.4 CO$_2$ flux

The $F_C$ module adopts a bottom-up approach to determine the local-scale $F_C$ ($\mu$mol m$^{-2}$ s$^{-1}$), accounting for both anthropogenic ($F_{C,ant}$) and biogenic ($F_{C,bio}$) components (Järvi et al., 2019):

$$F_C = F_{C,ant} + F_{C,bio} = (F_M + F_V + F_B + F_P) + (F_{pho} + F_{res}). \tag{10}$$

In the Eq. 10, $F_M$ are CO$_2$ emissions from human metabolism, $F_V$ emissions from traffic, $F_B$ emissions from buildings (e.g., combustion of natural gas, coal and wood), $F_P$ emissions from local-scale point sources, $F_{pho}$ photosynthesis, and $F_{res}$ vegetation and soil respiration. Positive values indicate sources of CO$_2$ and negative values sinks with respect to the atmosphere.

$F_{C,ant}$ relates to energy balance through $Q_F$. $F_M$ and $F_V$ are estimated based on an inventory approach, i.e., based on population density or traffic rate, and their emission factors (EFs). Hourly CO$_2$ emissions from human metabolism on workdays/weekends ($F_{M,h,d}$, $\mu$mol m$^{-2}$ s$^{-1}$) are calculated using:

$$F_{M,h,d} = p_{h,d} \cdot H_{a,h,d} \cdot C_M, \tag{11}$$

where $p_{h,d}$ is the population density (cap ha$^{-1}$), $H_{a,h,d}$ activity by hour calculated from the diurnal profiles for population and activity, and $C_M$ CO$_2$ released per person ($\mu$mol CO$_2$ s$^{-1}$ cap$^{-1}$). $C_M$ varies between nighttime minimum and daytime maximum values ($C_{M(min,max)}$).

Hourly traffic CO$_2$ emissions ($F_{V,h,d}$) on weekdays or weekends are calculated from

$$F_{V,h,d} = T_{rd} \cdot E_{c,d} \cdot H_{T,d}, \tag{12}$$





where $T_{rd}$ is the mean daily traffic rate within the study area (veh day$^{-1}$ area$^{-1}$), $H_{T,d}$ diurnal traffic profiles, and $E_{c,d}$ traffic EFs for $CO_2$ (kg km$^{-1}$ veh$^{-1}$).

Hourly building $CO_2$ emissions ($F_{B,h,d}$) on weekdays or weekends are calculated from

$$F_{B,h,d} = [fr_{heat} \cdot (Q_{F,heat} + Q_{F,cool}) + fr_{nonheat} \cdot Q_{F,base} \cdot fr_{QF,base,BEU,d}] \cdot E_{CO2perJ}, \tag{13}$$

where $fr_{heat}$ is the fraction of fossil fuels used for heating, $Q_{F,heat}$ building heat emission at local scale estimated from the heating-degree-day model (Järvi et al., 2011), $fr_{nonheat}$ fraction of fossil fuels used for energy other than heating (e.g. combustion from domestic cooking), $Q_{F,base}$ non-temperature related anthropogenic heat flux (W m$^{-2}$), $fr_{QF,base,BEU,d}$ the

fraction of the $Q_{F,base}$ coming from building energy use on weekdays or weekends, and $E_{CO2perJ}$ the EF for fuels in building energy use. Emissions from single-point sources such as power plants and industrial activities can be included as model inputs.

$F_{pho}$ relates to energy balance through LAI and the environmental responses of surface conductance (Eq. 3). $F_{pho}$ is calculated using

$$F_{pho} = \sum_i (fr_i F_{pho,max,i} LAI_i) g(T_{air}) g(\Delta q) g(\Delta \theta) g(K_{down}), \tag{14}$$

where $F_{pho,max,i}$ is the maximum photosynthetic rate for vegetation type $i$.

Soil and vegetation respiration $F_{res}$ ($\mu$mol m$^{-2}$ s$^{-1}$) follows an exponential dependency on $T_{air}$:

$$F_{res} = \sum_i fr_i \max(a_i \cdot \exp(T_{air} b_i), 0.6), \tag{15}$$

where $a_i$ and $b_i$ are parameters controlling the temperature dependency, and 0.6 $\mu$mol m$^{-2}$ s$^{-1}$ is the minimum respiration in winter time.

## 3   Study site and measurements

The model domain is located around a 325-m meteorological tower constructed by Institute of Atmospheric Physics, Chinese Academy of Sciences (IAP tower, 39°58' N, 116°22' E, 60 m above sea level) within the 6$^{th}$ Ring area of Beijing, China (Fig. 1 a). An EC setup at the height of 47 m on the IAP tower continuously measures the surface fluxes of $Q_E$, $Q_H$ and $F_C$ using a 3-dimensional sonic anemometer (Windmaster, Gill, UK) and an open-path infrared gas analyzer (LI-7500A, LI-COR,

USA). In addition, all four radiation components are measured at the height of 140 m using a net radiometer (CNR1, Kipp & Zonen, Netherlands). These measurements are used to evaluate SUEWS model performance. The 1 km radius circle around the IAP tower roughly covers 80% accumulated flux footprint area (Liu et al., 2012). This area is mainly covered by impervious surfaces (Fig. 1 b). Three patches of urban green spaces are situated to the east, south and west to the IAP tower, while the other vegetation is scarcely located along the roads and near the buildings. Most of the impervious surfaces in the source area

are residential buildings (Fig. 1 c). A more detailed description of the surroundings and the used instruments can be found in Liu et al. (2012), Liu et al. (2021) and Cheng et al. (2018).



**Figure 1.** (a) The location of IAP tower and the land cover type within the 6$^{th}$ Ring area of Beijing (Terra and Aqua combined Moderate Resolution Imaging Spectroradiometer (MODIS) Land Cover Type (MCD12Q1) Version 6) (Friedl and Sulla-Menashe, 2019), (b) a satellite image (Google Earth, image ©2022 Maxar Technologies) and (c) urban land use categories (EULUC-China) (Gong et al., 2020) covering the circle of 1 km radius around the IAP tower.

The following quality control steps are performed for 30-minute turbulent flux observations in the year 2016. (1) Upper/lower boundary filtering: $Q_E$ observations that fall outside the range from -500 to 1000 W m$^{-2}$, $Q_H$ from -500 to 1000 W m$^{-2}$, and $F_C$ from -100 to 200 $\mu$mol m$^{-2}$ s$^{-1}$ are removed. (2) Spikes detection: flux values outside 3.5 times of the standard deviation from the 3-day moving mean value are removed (Liu et al., 2012). (3) Wind direction filtering: the wind directions






with building heights over 50 m (112–128, 160–243, 314–3°) are removed (Kokkonen et al., 2019). (4) Stationarity test: data points with stationary indicator > 30% are filtered out (Foken and Wichura, 1996). The percentages of data removed through these 4 steps are 0.2–0.3%, 1.7–2.7%, 37.8–38.2% and 13.1–17.4%, respectively. The numbers of observations retained after quality control are 8017 for $Q_E$, 7338 for $Q_H$ and 7797 for $F_C$. These 30-minute flux observations are resampled to one–hour

resolution. In addition, $F_C$ is gap-filled using seasonal mean diurnal cycle in order to calculate the seasonal and annual sums (Falge et al., 2001).

To optimize the behaviour of LAI with Covariance matric adaptation evolution strategy (CMA-ES) (Appendix A), nine-year time series (2008–2016) of LAI within urban area of Beijing (the 6th Ring area) are derived from the MOD15A2H Version 6 Moderate Resolution Imaging Spectroradiometer (MODIS) combined Leaf Area Index (LAI) and Fraction of Photosyn-

thetically Active Radiation (FPAR) product (Myneni et al., 2015). Related data and codes are openly available (Zheng et al., 2022).

## 4 Model run

### 4.1 Forcing meteorological data

The reanalysis dataset WFDE5 (Cucchi et al., 2021) is used as the forcing data for SUEWS. WFDE5 is a bias-corrected dataset

of near-surface meteorological variables specifically suited for land surface modelling. It is derived from the fifth generation of the European Centre for Medium-Range Weather Forecasts (ECMWF) atmospheric reanalysis (Hersbach et al., 2020). It is provided at 0.5° spatial and at hourly temporal resolution. WFDE5 is evaluated against observed meteorological observations before it is used as the forcing data for SUEWS (Appendix B).

### 4.2 Land cover

Land cover types and their fractions needed in the model runs are estimated based on aerial images. Paved surface accounts for 46% of the total area, buildings 24%, trees/shrub 13%, grass/lawn 16%, and water 1%. The average building height is 19.1 m (Kokkonen et al., 2019). According to a field survey conducted in the 6th Ring area, the population of deciduous species accounts for 82% of the total number of woody plants investigated (Ma, 2019). Therefore, the surface fraction of deciduous trees is set to 11% and evergreen trees 2%.

To calculate the storage heat flux, Objective Hysteresis Model (OHM) is used (Grimmond and Oke, 1999), and the coefficients for paved surface is set to the weighted average of case AN99 to represent the asphalt surface following Ward et al. (2016).

### 4.3 Human activity

In this study, local parameters of traffic, population dynamics, and building energy use are incorporated when compared to

Kokkonen et al. (2019) as $CO_2$ emissions are more sensitive to them when compared to anthropogenic heat flux.


Traffic volume and fleet mix data for each link of the study domain for the year 2017 are obtained from Yang et al. (2019). Hourly traffic rate on weekdays or weekends is calculated as the sum of traffic volume weighted by road length within the study domain, and mean daily traffic rates ($Tr_{wd/we}$) and their diurnal profiles are then obtained (Table 1, Fig. 2). The average traffic $CO_2$ EFs are calculated using the method following Zhang et al. (2018) based on the self-reported fuel records, on-road

measurements, vehicle category and speed obtained in Beijing (Zhang et al., 2014; Wen et al., 2020, 2022) (Table 1). Traffic rate during daytime is higher than in the nighttime, and it is higher on weekdays than on weekends throughout the day (Fig. 2).

Annual average diurnal cycle of population density within the model domain is obtained from a dataset of hourly population dynamics at 500-m resolution generated from remotely sensed and geospatial data over years 2015 and 2016 (Zhao et al., 2021) (Fig. 2). As there are several residential building areas located within the model domain (Fig. 1 c), population density increases

in the evening when residents get home from work, remains high throughout the night, and decreases in the morning.

Heating in Beijing is dominated by central heating, supplied mainly at the district level. The sources include cogeneration plants fueled by coal or gas, and district boilers powered by coal, oil, or gas. In 2015, the ratio of district boilers heating capacity to cogeneration plants was 4.2:1 (Zhang et al., 2019). Cogeneration plants are usually located at sub-urban or rural areas, and there are no cogeneration plants within the model domain, so their contribution to $CO_2$ emission is neglected in this study. In

comparison, over 5000 coal-fired and 1000 gas-fired heating boilers are located surrounding the populated areas in 2014 (Cui et al., 2019). The exact number or heating capacity is unknown in the study area, but three boiler plants are seen within 2 km distance from the IAP tower through Baidu Map. Their $CO_2$ emissions are very likely to be observed by EC during heating season. Therefore, fraction of fossil fuels used for heating ($Fr_{heat}$) is set to 0.81 based on the ratio of heating capacity. SUEWS first estimates the anthropogenic heat, then converts the heat into $CO_2$ combustion using the EF and emission fractions. The

value of natural gas consumption over the annual total heating supply is $3.2 \times 10^6$ ton coal equivalent (tce) and the value for coal consumption $2.6 \times 10^6$ tce in 2015. The consumption of rest of the fuel types is only $3.8 \times 10^5$ tce (MHURD, 2018). The EFs of heating supply are 96.51 Tg $CO_2/10^{19}$ J for coal-fired boiler, and 56.17 Tg $CO_2/10^{19}$J for gas-fired boiler, respectively (Du et al., 2018). Therefore, the EF for fuels used in building energy use ($E_{CO2perJ}$) in SUEWS takes the average of the EFs of natural gas and coal weighted by their annual consumption, i.e., 0.1688 $\mu$mol $CO_2$ J$^{-1}$ (Table 1). In addition, SUEWS needs a

temperature limit (base temperature, $T_{base\_HC}$) to indicate when heating takes place in the heating-degree-day model. Central heating usually starts around 15[th] of November and lasts until 15[th] of March, when the outdoor air temperature is around 12 °C. Therefore, this value is given to SUEWS $T_{base\_HC}$ to replace the default value (18.2 °C) (Järvi et al., 2011).

Household fuel combustion, mainly from cooking, takes place throughout the year, but its $CO_2$ emission are relatively small. An observational study shows that $CO_2$ emitted from fuel combustion in household cooking contributes only 6% of the indoor

$CO_2$, and this percentage is low, compared with contribution by human metabolism (30%) (Shen et al., 2020). Therefore, we set 20% to non-heating energy fraction ($Fr_{nonheat}$).





**Table 1.** Parameters related to anthropogenic heat and $CO_2$ emissions.

| Parameter | Notation | Value | Reference |
|---|---|---|---|
| $C_{M(min)}$ | Minimum $CO_2$ release per capita | 120 $\mu$mol $CO_2$ s$^{-1}$ cap$^{-1}$ | Ward et al. (2013) |
| $C_{M(max)}$ | Maximum $CO_2$ release per capita | 280 $\mu$mol $CO_2$ s$^{-1}$ cap$^{-1}$ | Moriwaki and Kanda (2004) |
| $E_{c,wd}$ | Traffic $CO_2$ EF for weekday | 0.207 kg km$^{-1}$ | Zhang et al. (2014); Wen et al. (2020, 2022) |
| $E_{c,we}$ | Traffic $CO_2$ EF for weekend | 0.209 kg km$^{-1}$ | Zhang et al. (2014); Wen et al. (2020, 2022) |
| $Tr_{wd}$ | Mean daily traffic rate for weekday | 0.3260 km day$^{-1}$ m$^{-2}$ | Yang et al. (2019) |
| $Tr_{we}$ | Mean daily traffic rate for weekend | 0.2664 km day$^{-1}$ m$^{-2}$ | Yang et al. (2019) |
| $Fr_{heat}$ | Fraction of fossil fuels used for heating | 0.81 | Cui et al. (2019); Zhang et al. (2019); MHURD (2018) |
| $Fr_{nonheat}$ | Fraction of fossil fuels used for energy | 0.2 | this study |
| $E_{CO2perJ}$ | EF for fuels used in building energy use | 0.1688 $\mu$mol $CO_2$ J$^{-1}$ | Cui et al. (2019) |
| $T_{base\_HC}$ | Base temperature for heating degree day | 12 °C | this study |

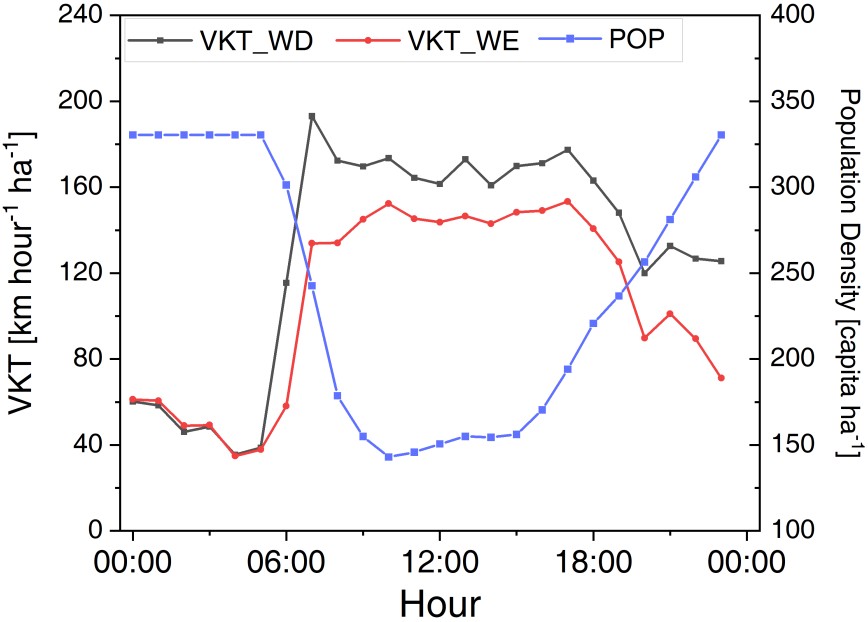

**Figure 2.** Annual average diurnal cycle of traffic rate (vehicle kilometers travelled, VKT) for weekday (VKT_WD), weekend (VKT_WE), and population density (POP) within the 1 km radius circle around the IAP tower.





## 4.4 Evaluation design

Two different groups of SUEWS runs are made around the IAP tower (Fig. 1c). The first run over years 2009 to 2011 is to evaluate the modelled radiation components against observations. The first 16 months are the spin-up period, and the actual
model evaluation are made from May 2010 to June 2011, when the radiation observations are available. The second SUEWS run from 2015–2016 is to evaluate the turbulent fluxes. The first year is used as a spin-up period and only year 2016 is used in the evaluation.

The calculation of radiation fluxes is mostly dependent on the land cover fractions under the current scheme adopted by SUEWS (Loridan et al., 2011; Offerle et al., 2003). No visible change in land use type is observed according to satellite
images from Google Earth within the modelled area between 2010 and 2016 (figure not shown). Therefore, we assume that the evaluation of radiation fluxes using observations in years 2011–2012 holds true in 2016.

### 4.4.1 Sensitivity to vegetation-related parameters

In order to test the model's sensitivity of radiation and turbulent fluxes to vegetation-related parameters, four model runs are designed as follows:

1. Case **base**: control run where model parameters are considered "default" following Kokkonen et al. (2019) (Table S1–S3). The exceptions are: (1) parameters in the environmental response functions (Eq.4 –7) of surface conductance which follow those by Havu et al. (2022a) (Table 2), because the product of response functions calculated following Kokkonen et al. (2019) is too low (95[th] percentile = 0.19) to obtain realistic estimate of photosynthetic rate; (2) biogenic parameters for Eq.14–15 and soil properties which were updated following Havu et al. (2022a) (Table 2). In addition, $F_{pho,max}$
for grass/lawn (5.5 $\mu$mol m$^{-2}$ s$^{-1}$) is obtained from EC observations of $CO_2$ flux made over urban lawn in Helsinki in summer 2021 by fitting the conductance parameters ($F_{pho,max,grass}$, $G_2 - G_6$) to the observations following Järvi et al. (2019) (Appendix C).

2. Case **LAI**: Same as the case **base** but the parameters for Eq.1 describing the annual behaviour of LAI are optimized using remotely-sensed LAI and CMA-ES (Appendix A). The new optimized LAI parameters are compared with case
**base** in Table 4.

3. Case **gs**: Same as the case **base** but with $g_{max}$ values collected from observational studies over vegetation species in Beijing (Appendix D). The site-specific $g_{max}$ are in general lower than the values used by case **base** (Table 3).

4. Case **gs_LAI**: Same as the case **base**, but with both LAI and $g_{max}$ modified as described in case **LAI** and case **gs**.





**Table 2.** SUEWS biogenic model parameters used for all case runs in this study.

| Parameter | Evergreen/deciduous tree | Grass/lawn | Reference |
|---|---|---|---|
| $LAI_{i,max}$ (m$^2$ m$^{-2}$) | 5.1[a]/5.5[b] | 5.9 | Järvi et al. (2011) |
| $LAI_{i,min}$ (m$^2$ m$^{-2}$) | 4.0[a]/1.0[b] | 1.6 | Järvi et al. (2011) |
| $F_{pho,max,i}$ ($\mu$mol m$^{-2}$ s$^{-1}$) | 8.4[c] | 5.5[d] | [c]Havu et al. (2022a); [d]this study |
| $G_1$ | 3.5 | 3.5 | Havu et al. (2022a) |
| $G_2$ | 477 | 477 | Havu et al. (2022a) |
| $G_3$ | 0.66 | 0.66 | Havu et al. (2022a) |
| $G_4$ | 0.89 | 0.89 | Havu et al. (2022a) |
| $G_5$ | 30 | 30 | Havu et al. (2022a) |
| $G_6$ | 0.36 | 0.36 | Havu et al. (2022a) |
| $\Delta\theta_{WP}$ (mm) | 132 | 132 | Havu et al. (2022a) |
| $K_{down,max}$ (W m$^{-2}$) | 1200 | 1200 | Järvi et al. (2014) |
| $T_L$ (°C) | -10 | -10 | Ward et al. (2016) |
| $T_H$ (°C) | 55 | 55 | Ward et al. (2016) |
| $a_i$ | 0.78[c] | 2.1[e] | [c]Havu et al. (2022a); [e]Järvi et al. (2019) |
| $b_i$ | 0.08[c] | 0.06[e] | [c]Havu et al. (2022a); [e]Järvi et al. (2019) |
| Soil depth (mm) | 1000[c] | 349[f] | [c]Havu et al. (2022a); [f]Kokkonen et al. (2019) |

[a] Evergreen tree

[b] Deciduous tree

**Table 3.** Comparison of maximum conductances ($g_{max}$) between the case **base** and case **gs/gs_LAI**. Parameters for case **base** follow Järvi et al. (2011). More details can be found in Appendix D.

| | $g_{max}$ (mm s$^{-1}$) | |
|---|---|---|
| | case **base** | case **gs/gs_LAI** |
| Evergreen tree | 7.4 | 1.4 |
| Deciduous tree | 11.7 | 7.0 |
| Grass | 40.0 | 3.7 |

### 4.4.2 Sensitivity to radius of modelled area

SUEWS output will be evaluated against EC measured turbulent fluxes, but the challenge is that the source area of the observations are different to the exact modelling domain. To consider the impact of the chosen modelling domain on model evaluation, we designed three additional model runs where different radius circular areas around the IAP tower are considered. The default run is with the 1 km radius circle, but we also run SUEWS within 500 m, 750 m, and 1500 m circular areas, corresponding to





**Table 4.** Comparison in leaf area index (LAI) parameters between case **base** and case **LAI/gs_LAI**. All vegetation types (evergreen tree, deciduous tree and grass) use the same LAI parameters within one case. Case **base** values are as in Järvi et al. (2011).

|  | LAI parameters | |
| --- | --- | --- |
|  | case **base** | case **LAI/gs_LAI** |
| $T_{base,GDD}$ ($^{\circ}$C) | 5 | 3.8 |
| $T_{base,SDD}$ ($^{\circ}$C) | 10 | 35 |
| $GDD_{full}$ ($^{\circ}$C) | 300 | 2235 |
| $SDD_{full}$ ($^{\circ}$C) | -450 | -2365 |
| $\omega_{1,GDD}$ | 0.04 | -1.85 |
| $\omega_{2,GDD}$ | 0.001 | 0.00029 |
| $\omega_{1,SDD}$ | -1.5 | 1.28 |
| $\omega_{2,SDD}$ | 0.0015 | 0.000025 |

the flux footprint of roughly 60%, 70%, and 80–90%, respectively (Liu et al., 2012). Used vegetation parameters are as in case
**gs_LAI**, but land surface fractions, population, and traffic parameters are modified accordingly (Appendix E).

### 4.5 Statistical metrics for model evaluation

Common statistical metrics are used to quantify the model performance, including, coefficient of determination ($R^2$), root-mean-square error (RMSE) and mean bias error (MBE). Simple linear regression is used to estimate the relationship between the model output and the observations, and the square of the correlation coefficient is taken as $R^2$. The other statistical metrics
are defined as follows:

$$\text{RMSE} = \sqrt{\frac{\sum_{i=1}^{n}(y_i - \hat{y_i})^2}{n}}, \text{and} \tag{16}$$

$$\text{MBE} = \frac{1}{n}\sum_{i=1}^{n}(\hat{y_i} - y_i), \tag{17}$$

where $\hat{y_i}$ is the modelled and $y_i$ the measured value. Statistical metrics are calculated at annual and seasonal scale, i.e., DJF
(winter), MAM (spring), JJA (summer), SON (autumn).

### 5 Results and Discussion

### 5.1 Seasonal dynamics of optimized LAI

The case **base** has the ability to simulate the onset of leaf growth and the ending of senescence well, but the general vegetation phenology modelling performs poorly (Fig. 3). The performance of LAI modelling is remarkably good after the optimization,





with high R$^2$ (0.94) and low RMSE (0.4 m$^2$ m$^{-2}$) (Appendix A). In the control case **base**, modelled LAI starts to rapidly increase from day of year (DOY) 70 and plateaus at DOY 105, which is too early when compared to remotely sensed LAI (MODIS LAI). Optimized LAI starts to grow at the same time but much slower and peaks around DOY 200. In autumn, LAI modelled by case **base** drops rapidly at DOY 310, while optimized LAI starts to decline steadily soon after its peak. LAI model with optimized parameters is better at capturing the behaviours of growth, peak growing season, and senescence for the year

2016 than in the control case **base**.

Although previous studies suggest that LAI is generally modelled well using default parameters following Järvi et al. (2011), Ward et al. (2016) reported that leaf-on is reached too early and suddenly in spring in two UK cities. In the contrary, Ao et al. (2018) showed that the simulated LAI might be lower than reality when vegetation remains green in winter and spring in Shanghai, China. These lead to the bias in $g_s$ and therefore $Q_E$ estimates (Eq. 2–3). In Beijing, the rainy season lasts from

May to October, while the other time of the year is the dry season (Liu et al., 2012). This influences the LAI behaviour such as the growth and senescence rates due to the lack of soil moisture in spring and autumn (Omidvar et al., 2022). Therefore, it is necessary to evaluate the LAI model when applied to a different city, and optimization of the LAI model is recommended when the LAI model performs poorly.

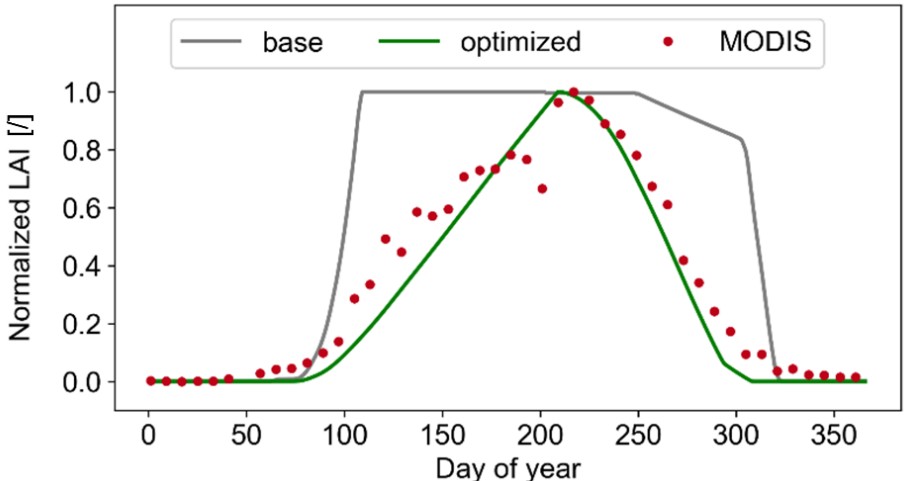

**Figure 3.** Normalized LAI in 2016, where the value 0 (1) represents the minimum (maximum) of LAI. "Base" denotes the modelled LAI for the case **base**, "MODIS" the remotely sensed LAI within the 6$^{th}$ Ring area of Beijing, and "optimized" the modelled LAI using the parameters derived with CMA-ES (Appendix A).

## 5.2  Evaluation of radiation fluxes

Four model experiments are conducted to examine the impact of $g_{max}$ and LAI seasonal dynamics on the modelled radiation fluxes (See Sect. 4.4.1). However, the choice of $g_{max}$ and LAI parameters has only a minor impact on the modelled radiation fluxes ($K_{down}$, $K_{up}$, $L_{down}$, $L_{up}$). For all of them, the R$^2$ between the case **gs_LAI** and the other cases is 1, and the absolute





values of RMSE and MBE are lower than 0.002 W m$^{-2}$ and $10^{-4}$ W m$^{-2}$, respectively. Therefore, only the case **gs_LAI** is further analyzed in this section.

300 The model performance of SUEWS in simulating radiation fluxes is good (Fig. 4, Table 5) and it can reproduce the diurnal cycle of each radiation flux well (Fig. 5). $K_{up}$ is overestimated in all seasons with MBE ranging from 0 to 10 W m$^{-2}$. At least partly, this overestimation can be explained by the positive bias (MBE $>$ 15 W m$^{-2}$ in all seasons) of WFDE5 $K_{down}$ when compared to the observed $K_{down}$ (Appendix B). The overestimation in $K_{up}$ can also be caused by surface albedo. Observational studies have shown that the urban surface albedo near the IAP tower varies between 0.1 and 0.15 with season

305 (Jiang et al., 2007; Miao et al., 2012). The annual average albedo for modelling domain given to SUEWS is 0.14, which is relatively high but still consistent with the observations. Larger positive bias is observed in summer than in winter. In SUEWS, albedo is allowed to vary between summer and winter if specified, and this can help to lower the bias in summer when the albedo for vegetation is low, but this is likely to have only a minor impact because the vegetated fraction is low.



**Figure 4.** Input or modelled vs. observed hourly radiation fluxes, including (a–d) incoming solar radiation ($K_{down}$), (e–h) outgoing shortwave radiation ($K_{up}$), (i–l) incoming longwave radiation ($L_{up}$), (m–p) outgoing longwave radiation ($L_{up}$), and (q–t) net radiation ($Q_N$) from May 2010 to June 2011. Note that only $K_{down}$ is input and the rest are model output.





**Figure 5.** Average diurnal cycle of input or modelled and observed hourly radiation fluxes by season, including (a–d) incoming solar radiation ($K_{down}$), (e–h) outgoing shortwave radiation ($K_{up}$), (i–l) incoming longwave radiation ($L_{up}$), (m–p) outgoing longwave radiation ($L_{up}$), and (q–t) net radiation ($Q_N$) from May 2010 to June 2011. Shading area denotes the standard deviation. Note that only $K_{down}$ is input and the rest are model output.

The average seasonal and diurnal cycles of $L_{down}$ are well captured by the model (Fig. 4 i–l), although $R^2$ (0.68–0.88) is
lower than with other radiation fluxes. The difference might result from the discrepancy between the observed and modelled cloud fraction as clear skies and overcast conditions are especially difficult to capture by the radiation model NARP, as reported





also by Ao et al. (2016) and Ward et al. (2016). The diurnal amplitude of $L_{up}$ is slightly overestimated by SUEWS (Fig. 5 m–p). The model tends to overestimate $L_{up}$ particularly at high values of $L_{up}$ (Fig. 4 m–p). The values of emissivity of building materials used in SUEWS might be slightly lower than in reality.

We conclude that SUEWS is applicable to provide realistic estimate of radiation fluxes in Beijing, in general accordance with previous studies (Järvi et al., 2014; Karsisto et al., 2016; Ward et al., 2016), despite the absence of site-specific parameters.

**Table 5.** SUEWS model performance statistics for radiation fluxes. The abbreviations are the same as Fig. 4. Note that $K_{down}$ is an input and others are output of SUEWS.

|              | Season | $R^2$ | RMSE  | MBE   | N    |
|--------------|--------|-------|-------|-------|------|
| $K_{down}$   | DJF    | 0.94  | 52.3  | 19.5  | 2160 |
|              | MAM    | 0.93  | 82.6  | 19.8  | 2940 |
|              | JJA    | 0.86  | 110.4 | 39.3  | 2728 |
|              | SON    | 0.93  | 59.6  | 16.6  | 2150 |
| $K_{up}$     | DJF    | 0.88  | 9.2   | 0.2   | 2160 |
|              | MAM    | 0.92  | 14.8  | 5.0   | 2940 |
|              | JJA    | 0.88  | 18.0  | 8.1   | 2728 |
|              | SON    | 0.91  | 11.6  | 3.3   | 2150 |
| $L_{down}$   | DJF    | 0.74  | 16.0  | -2.4  | 2160 |
|              | MAM    | 0.86  | 20.5  | -10.0 | 2940 |
|              | JJA    | 0.68  | 18.1  | 8.8   | 2728 |
|              | SON    | 0.88  | 23.7  | 12.9  | 2150 |
| $L_{up}$     | DJF    | 0.80  | 15.3  | 3.8   | 2160 |
|              | MAM    | 0.90  | 18.6  | 3.2   | 2940 |
|              | JJA    | 0.79  | 22.1  | 10.5  | 2728 |
|              | SON    | 0.91  | 18.6  | 9.2   | 2150 |
| $Q_N$        | DJF    | 0.94  | 38.0  | 13.2  | 2160 |
|              | MAM    | 0.93  | 64.6  | 1.5   | 2940 |
|              | JJA    | 0.86  | 87.0  | 29.5  | 2728 |
|              | SON    | 0.92  | 50.8  | 17.0  | 2150 |





### 5.3 Evaluation of turbulent heat fluxes modelling

Both observed $Q_E$ and $Q_H$ reach their maxima around noon (Fig. 6). The observed $Q_E$ has the largest amplitude during the summer months (JJA), while $Q_H$ during the spring months (MAM). All four model runs capture their diurnal cycles, but large

differences in the amplitude and model performance are observed among the model runs (Fig. 7).

In the case **base**, the model overestimates $Q_E$ (with MBE from -7.4 to 48.6 W m$^{-2}$) except in winter months (DJF) (Fig. 7). With the optimized LAI (case **LAI**), slightly better model performance in $Q_E$ is seen with lower RMSE (11.5–91.2 W m$^{-2}$) and larger R$^2$ (0.20–0.58) when compared to case **base** (with RMSE 11.7–96.1 W m$^{-2}$ and R$^2$ 0.20–0.51). (Fig. 7 a–c). $Q_E$ is to a large extent determined by surface conductance which for the modelled area is scaled by $g_{max}$ of each vegetated surface

(Eq. 3). Clear improvement is observed when local $g_{max}$ is used (case **gs**), especially during summer, when the overestimation of $Q_E$ is largely reduced, RMSE drops to 11.3–54.7 W m$^{-2}$, and R$^2$ increases to 0.25–0.61. The introduction of local $g_{max}$ improves the model performance more than the optimized LAI. With both optimized LAI and local $g_{max}$ introduced (case **gs_LAI**), the R$^2$ and RMSE are similar to those at case **gs**, while MBE drops below 0 in each season, indicating the $Q_E$ is slightly underestimated throughout the year. The underestimation can be explained by the missing of combustion-derived water

vapor in SUEWS as observational studies have shown that combustion-derived water vapor often contributes 5–10% of total urban humidity during heating season (Fiorella et al., 2018; Liu et al., 2022; Salmon et al., 2017). The observed $Q_E$ ranges from 0 to 30 W m$^{-2}$ in winter when vegetation is dormant; this suggests that combustion and evaporation, as the dominant sources of $Q_E$, might lead to $Q_E$ at this magnitude.



**Figure 6.** Annual and seasonal mean diurnal cycles of observed and modelled (a–e) latent heat flux ($Q_E$) and (f–j) sensible heat flux ($Q_H$) for the four model runs (case **base**, **gs**, **LAI**, and **gs_LAI**) in the year 2016.





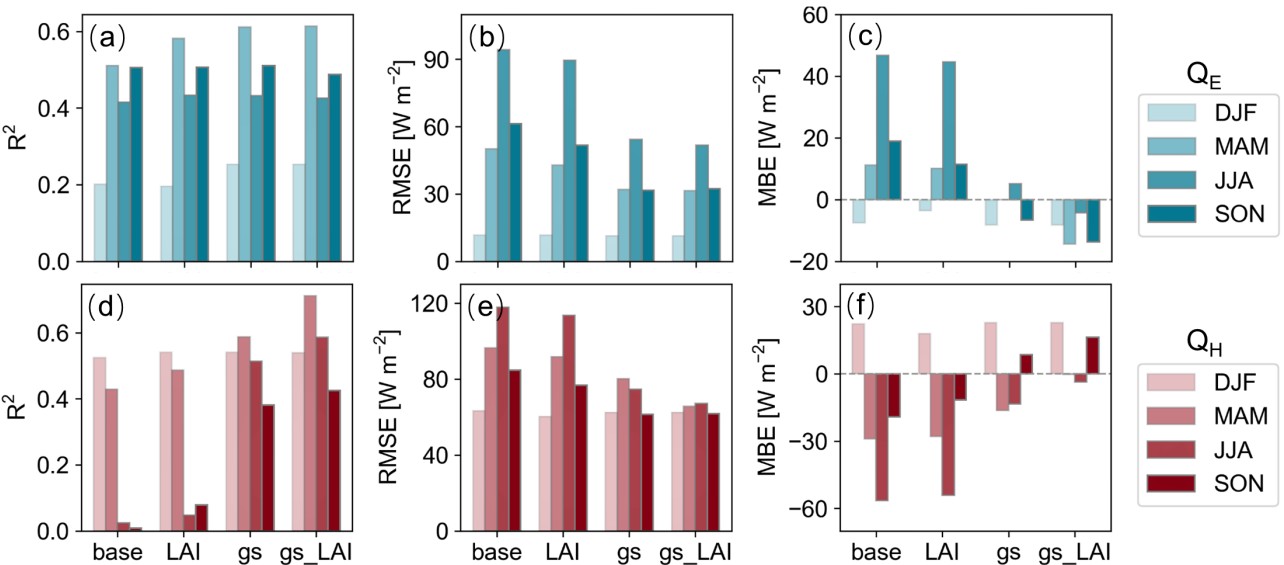

**Figure 7.** Model performance statistics $R^2$, RMSE and MBE of (a–c) latent heat flux ($Q_E$) and (d–f) sensible heat flux ($Q_H$) for the four model runs (case **base**, **gs**, **LAI**, and **gs_LAI**) in the year 2016.

In SUEWS, $Q_H$ is estimated as the residual of energy balance, and is therefore directly affected by the modelled $Q_E$. As a result of overestimating $Q_E$ in case **base**, $Q_H$ is greatly underestimated with MBE -51.1–14.4 W m$^{-2}$ and RMSE 60.9–116.0 W m$^{-2}$. $R^2$ in summer months and autumn months (SON) are lower than 0.1. The model performance is barely improved by optimized LAI (case **LAI**), but is noticeably improved after local $g_{max}$ is introduced (case **gs**) (Fig. 7 d–f). The best model performance for $Q_H$ is obtained by case **gs_LAI**, decreasing the magnitudes of MBE to -7.6–8.9 W m$^{-2}$ and RMSE to 58.0–69.1 W m$^{-2}$, and increasing $R^2$ to 0.42–0.72 (Fig. 7 d–e).

Our results suggest that both altering the values of $g_{max}$ based on observations and LAI model optimization can serve as effective approaches to improving heat fluxes modelling.

### 5.4 Evaluation of CO$_2$ fluxes modelling

#### 5.4.1 Model performance

SUEWS reproduces the average annual and seasonal diurnal cycle of observed $F_C$ (Fig. 8). The diurnal behaviour is dominated by on-road traffic emission, which reaches 22.6 and 23.0 $\mu$mol m$^{-2}$ s$^{-1}$ for the morning peak and afternoon peak, respectively, during the rush hours (Fig. 9). Human metabolism (maximum 4.8 $\mu$mol m$^{-2}$ s$^{-1}$) is the second largest source of CO$_2$ emissions. In winter, the building CO$_2$ emission has a maximum of 5.3 $\mu$mol m$^{-2}$ s$^{-1}$ in the daytime. The maximum photosynthesis rate is 5.0 $\mu$mol m$^{-2}$ s$^{-1}$ around noon in summer, while soil and vegetation respiration constantly serves as a CO$_2$ source with a rate lower than 3.6 $\mu$mol m$^{-2}$ s$^{-1}$. Each of the model performance statistics of $F_C$ is of a similar magnitude among cases, indicating the $F_C$ modelling is far less sensitive to the choice of $g_{max}$ and LAI-related parameters than $Q_E$ and





$Q_H$ shown in Sect. 5.3 (Fig. 10). In SUEWS, photosynthetic rate and respiration are multiplied by the percentages of vegetated surfaces, which account for only 29% of the modelled area. The magnitude of photosynthetic rate is substantially lower than the traffic emission, making the effect of photosynthesis hardly visible in $F_C$ diurnal cycle.

Expectedly, SUEWS has difficulty in capturing the hourly variability of $F_C$, resulting in the overall low $R^2$ (0.16–0.21) and
high RMSE (12.6–16.4 $\mu$mol m$^{-2}$ s$^{-1}$) (Fig. 10 a). On one hand, observed $F_C$ has great variability at hourly scale throughout the year indicated by the large deviation (Fig. 8). On the other, under the current parameterization, two of the anthropogenic $F_C$ components are static: modelled traffic emission diurnal cycle is only dependent on whether it is weekend or weekday, and modelled human metabolism diurnal cycle is invariable throughout the year (Fig. 9), making it difficult to capture the extreme values. Other urban $F_C$ bottom-up modelling studies also reported similar challenges in modelling $F_C$ hourly variability in
Basel and in Helsinki (Stagakis et al., 2022; Järvi et al., 2019).

SUEWS gives a reasonable estimate of annual accumulated $F_C$ (8.4 kg C m$^{-2}$ yr$^{-1}$), which is 12% higher than the observed gap-filled value (7.5 kg C m$^{-2}$ yr$^{-1}$). $F_C$ is overall slightly underestimated in winter months (with MBE around -0.7 $\mu$mol m$^{-2}$ s$^{-1}$) while overestimated in the other seasons, with the largest MBE (4.6–5.5 $\mu$mol m$^{-2}$ s$^{-1}$) in summer (Fig. 10 c). SUEWS overestimates summer $F_C$, which might be explained by the underestimation of vegetation photosynthetic rate due to
the lack of site-specific parameters.

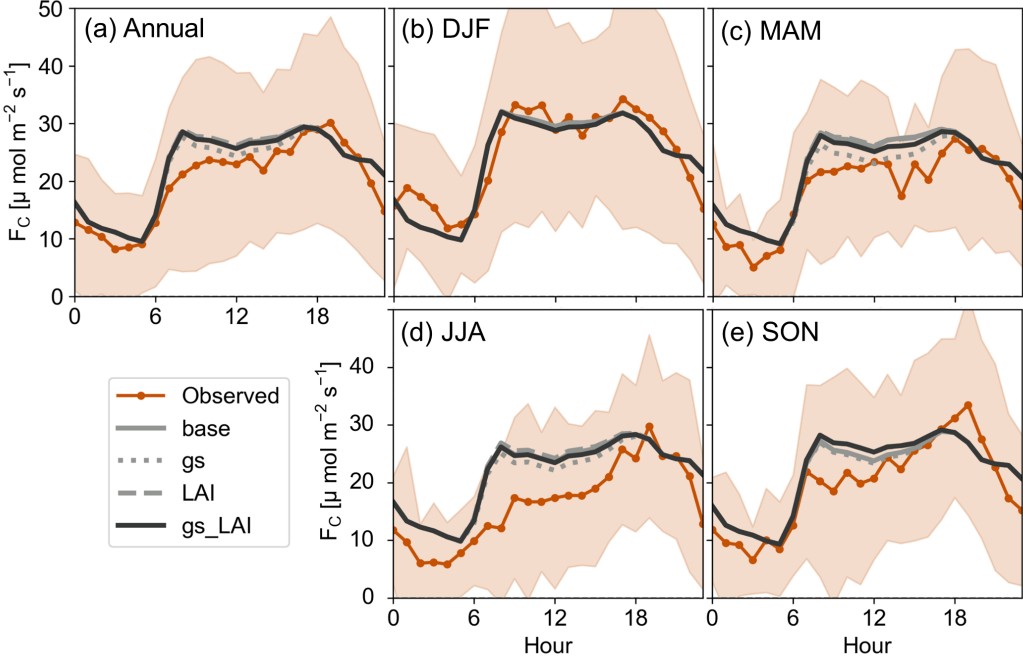

**Figure 8.** Annual and seasonal average diurnal cycles of observed and modelled $CO_2$ flux ($F_C$) for the four model runs (case **base**, **gs**, **LAI**, and **gs_LAI**) in the year 2016.





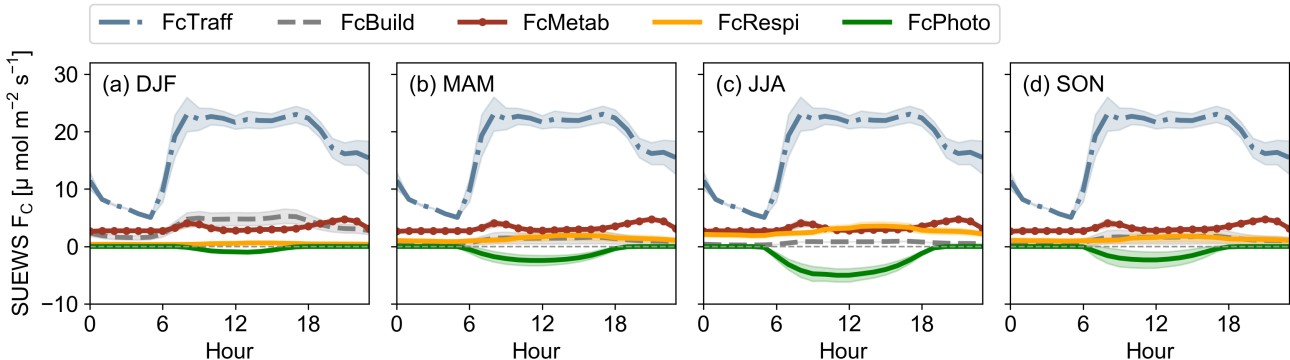

**Figure 9.** Seasonal average diurnal cycles of modelled $CO_2$ flux ($F_C$) components by case **gs_LAI** in 2016. FcTraff denotes $CO_2$ on-road traffic, FcBuilding building, FcMetab human metabolism, FcRespi vegetation and soil respiration, and FcPhoto vegetation photosynthesis. Positive values indicate sources of $CO_2$ and negative values sinks with respect to the atmosphere.

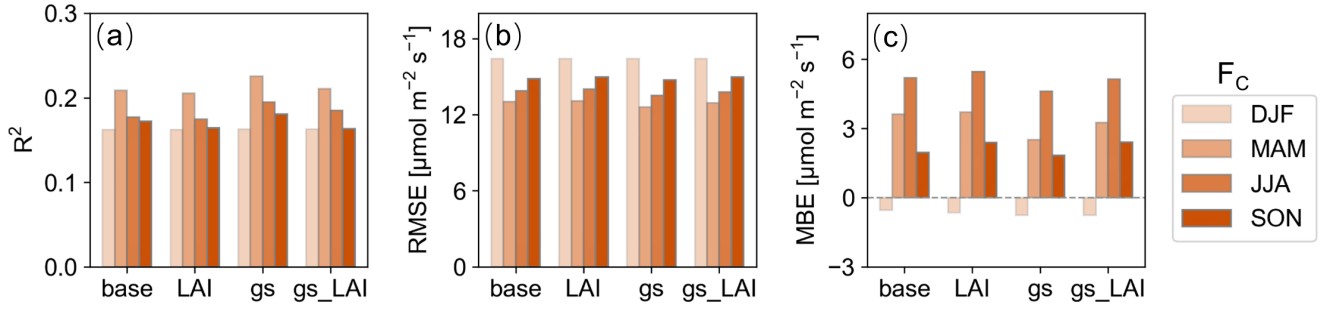

**Figure 10.** Model performance statistics (a) $R^2$, (b) RMSE and (c) MBE of $CO_2$ flux ($F_C$) for the four cases (**base**, **gs**, **LAI**, and **gs_LAI**) in the year 2016.

### 5.4.2 Model uncertainties

Uncertainties in traffic emission originate from the traffic rates and EFs. SUEWS adopts static traffic EFs, and neglects the relationship between traffic emission and $T_{air}$ as reported by Alvarez and Weilenmann (2012) and Fontaras et al. (2017). In order to examine the impact of seasonal variation of $T_{air}$ in traffic emission, correction is conducted using the regression function following Zhang et al. (2021), but only a marginal difference is seen at monthly scale: a difference of 3% in winter, -2% in spring, ~0% in summer and -1% in autumn. Therefore, we believe that the static traffic EFs adopted by SUEWS can provide reasonable traffic emission without considering the seasonal dynamics of $T_{air}$.

Järvi et al. (2019) reported that using a different coefficient of $CO_2$ release per capita ($C_M$) lead to a 6% decrease in human metabolic $CO_2$ emission estimate. If $C_M$ is set to a daily mean value of 242 $\mu$mol m$^{-2}$ s$^{-1}$ (Prairie and Duarte, 2007) instead



of the current values (Table 1), the human metabolic emission will increase and the annual $F_C$ will be 4% higher than the original estimate.

Building emission are calculated based on the $Q_F$ estimates and heating fraction. Modelled average $Q_F$ in December is 52.7 W m$^{-2}$, which is higher than another model estimate (21.6 W m$^{-2}$) in the modelled area (Wang et al., 2020). Observations of $Q_F$ are rarely available, and thus these $Q_F$ estimates have not yet been validated. The representativeness of the heating fraction

estimated from yearbook statistics is yet to be examined because the location and heating capacity of heating boilers within the modelled area is unknown. However, the building emission estimate (0.6 kg C m$^{-2}$ yr$^{-1}$) falls in the range of estimates (~0–3.0 kg C m$^{-2}$ yr$^{-1}$) by other cities (Björkegren and Grimmond, 2018; Järvi et al., 2019; Moriwaki and Kanda, 2004; Christen et al., 2011).

The modelled respiration is larger than $CO_2$ assimilated through photosynthesis in our study. At the annual scale, urban

vegetative surfaces can serve as a net $CO_2$ sink (Awal et al., 2010; Konopka et al., 2021), but may also vary from a net $CO_2$ sink to a source (Peters and McFadden, 2012). Admittedly, bias might exist in biogenic $CO_2$ flux estimates since the parameters used in this study are derived from the observations over street trees in Helsinki and over a lawn at Ossinlampi, Finland, where the climate and vegetative species are different from Beijing. With these parameters, the model might underestimate the $CO_2$ sequestrated by the local vegetation, and overestimate net $F_C$ in the growing season as shown in Sect. 5.4.1.

### 5.4.3   The impact of the modelling domain size

The surroundings of the IAP tower are heterogeneous in terms of land surface fraction and mean daily traffic rate (Fig. 11 a). The fraction of vegetated surfaces is higher closer to the tower than further away due to the green spaces adjoining the IAP tower (Fig. 1 b). Additionally, there is a traffic hot spot on the North 3$^{rd}$ Ring Road located 850 m to the south of IAP tower, where the traffic rate is 2 to 7 times the value for the other roads inside the circle of a 1000 m radius (figure not shown). A

large increase of 26% in daily traffic rate is seen when the radius of modelling domain is 1000 m or 1500 m when compared to domains with lower radii (Fig. 11 a). Thus, the modelled annual accumulated $F_C$ largely depends on the modelling domain size chosen, giving estimates of 7.2, 7.4, 8.4 and 8.5 kg C m$^{-2}$ yr$^{-1}$ for the radii of 500 m, 750 m, 1000 m, and 1500 m, respectively. Observational annual $F_C$ (7.5 kg C m$^{-2}$ yr$^{-1}$) falls within this range, which indicates the good model performance of SUEWS (Fig. 11 b).

Land surface fractions are critical parameters in turbulent flux modelling, but the use of site-specific fractions covering a certain area might not yield the best model performance (Demuzere et al., 2017; Loridan and Grimmond, 2012). Approximating the source area by choosing the radius of ≥80% footprint fetch (i.e. ≥1000 m in this study) does not give the closest estimate of annual $F_C$, showing agreement with previous studies. The model performance on heat and $F_C$ fluxes is reasonably good for each radius (figure not shown), but the selection of radius causes a difference up to 17.5% on annual $F_C$ estimate (Fig.

11b). This can be explained by the mismatch between the modelling domain and the real flux source area—the single fixed-extent modelled area cannot perfectly represent the land surface characteristics (e.g., the nonuniform land cover and human activities), biasing turbulent flux modelling (Chu et al., 2021; Laine et al., 2009). First, the accumulated footprint area of the observed fluxes is irregular in shape and vary with time (Liu et al., 2012), while many studies (Demuzere et al., 2017; Järvi





et al., 2019) including this study select a circular area to be modelled and evaluated. Changing the size of modelling domain
is challenging when soil processes are taken into account in SUEWS. Second, the relative contribution to flux from the land
surface decreases as the distance to the measurement instrument increases (Christen et al., 2011; Rebmann et al., 2005). When
the modelling domain is a 1000 m radius circle, the model might underestimate the relative contribution from the adjacent
vegetated surface, and overestimate the contribution of the traffic hot spot at the edge of 80% footprint fetch. We conclude that
the land surface model single-point evaluation needs to be performed with the awareness of the mismatch between flux source
area and modelling domain, especially over heterogeneous surfaces such as urban surfaces.

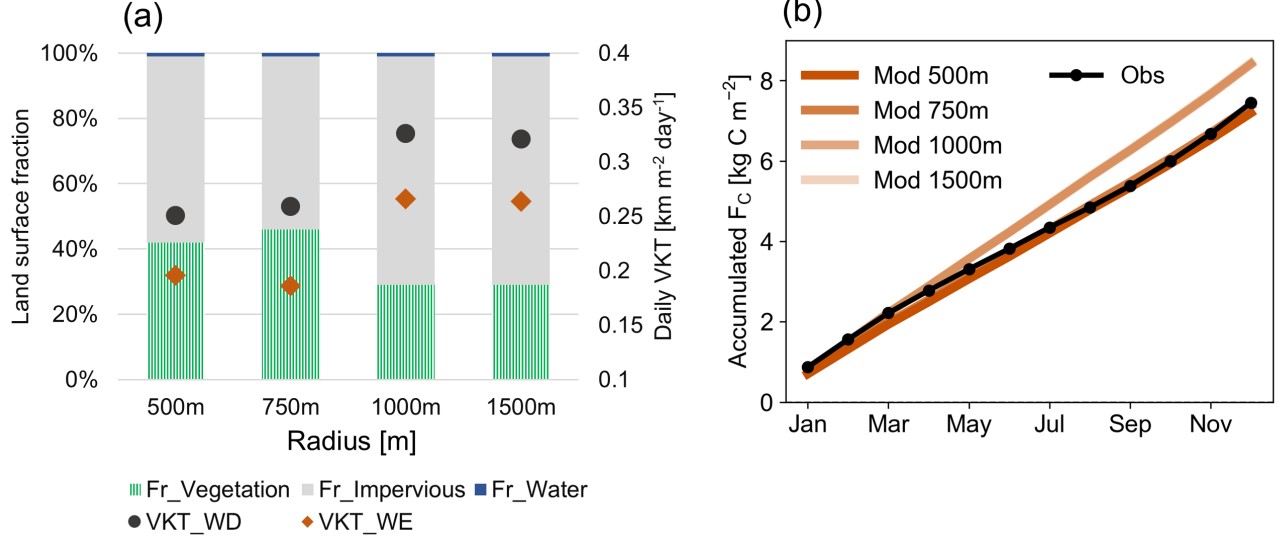

**Figure 11.** (a) Land use fraction and mean daily traffic rate (VKT, same as $Tr$ in Table 1), and (b) accumulated $CO_2$ flux ($F_C$) for modelling domain with the radius ranging from 500 m to 1500 m. "WD" denotes weekday, "WE" weekend; "Mod" denotes modelled $F_C$, and "Obs" observed $F_C$. Note that in (b) the lines for Mod 500m and Mod 750 m nearly overlap, and lines for Mod 1000m and Mod 1500m nearly overlap.

Regardless of the modelling domain size, traffic is the dominant $CO_2$ source contributing 63–73% to the total $CO_2$ emissions, followed by human metabolism (14–18%), respiration released by vegetation and soil (6–11%), and heating (6–9%). Vegetation photosynthesis offsets only 4–8% of the total annual $CO_2$ emissions (Fig. 12). Several bottom-up modelling studies show that on-road traffic is the greatest source in a densely built neighbourhood, contributing to 70% in central London, 61% in Helsinki,
53%–78% in Tokyo, 70% in Vancouver, while human metabolism also plays an important role, contributing 5–39% the annual total $F_C$ (Björkegren and Grimmond, 2018; Järvi et al., 2019; Moriwaki and Kanda, 2004; Christen et al., 2011). Our results are in general agreement with these studies. The contribution of building emission is more variable among cities: a contribution of 70% is reported in Basel (Stagakis et al., 2022), while ~0% in Helsinki (Järvi et al., 2019), and our estimate falls in this range. The direct $CO_2$ sequestration by urban plants is minor compared with the total $CO_2$ emissions in this densely built





neighborhood, which is in general agreement with Pataki et al. (2011) and Christen et al. (2011). For more accurate biogenic

component estimates in Beijing, photosynthetic and respiration observations over local species are needed in the future.

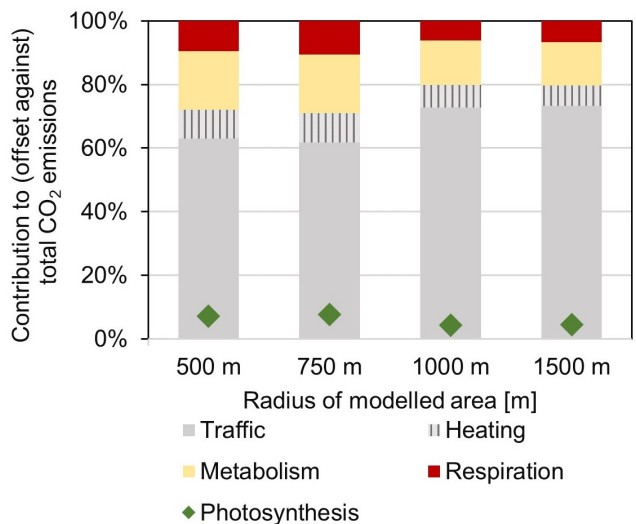

**Figure 12.** The contribution to total $CO_2$ emissions by each component at the annual scale for the modelled area with a radius ranging from 500 m to 1500 m. Note that for photosynthesis the percentages denote the offset against total $CO_2$ emissions.

## 6    Conclusions

A correct description of vegetation is vital in order to simulate the energy and $CO_2$ fluxes over urban surfaces using the urban

land surface model SUEWS. In this study, the impact of selecting appropriate vegetation parameters, including LAI parameters

and $g_{max}$, on simulating the surface fluxes is examined in Beijing, China. Besides, the newly-developed $CO_2$ emissions module

in SUEWS is evaluated against EC measurements.

Radiation flux modelling performs well without fine-tuning and it is hardly influenced by $g_{max}$ and LAI. The modelling

of heat fluxes including $Q_E$ and $Q_H$, however, shows great sensitivity to $g_{max}$ and the behaviour of LAI. LAI model using

"default" parameters from previous studies has difficulty in capturing the phenology dynamics (e.g., the rate of leaf-growth,

leaf-off) in Beijing, likely due to the dry wet season pattern. By using CMA-ES to optimize the LAI parameters with remotely

sensed LAI, the LAI modelling is noticeably improved ($R^2$=0.94, RMSE=0.4 m$^2$ m$^{-2}$). Observational leaf-level $g_{max}$ of

vegetative species in Beijing are also collected and parameterized. By incorporating local LAI and $g_{max}$, SUEWS simulated

the heat fluxes remarkably better, increasing $R^2$ by 0.02 (0.36) and decreasing RMSE by 27.4 (27.9) W m$^{-2}$ for $Q_E$ ($Q_H$),

and showing more realistic seasonal dynamics when compared to EC observations.

In comparison of heat flux modelling, $F_C$ modelling shows lower sensitivity to the choice of LAI-related parameters and

$g_{max}$. SUEWS can catch the general diurnal and seasonal behaviour of $F_C$ but tends to overestimate $F_C$, especially over

summer months. We also tested the influence of chosen modelling domain size on simulated $F_C$. By selecting the modelled





radii of circular area ranging from 500 m to 1500 m (i.e., accumulated footprint area from 60% to 80–90%), the modelled annual $F_C$ ranges from 7.2 to 8.5 kg C m$^{-2}$ yr$^{-1}$, which is comparable with the EC observations (7.5 kg C m$^{-2}$ yr$^{-1}$). This shows

the model performs well also on the annual scale. The variation in annual cumulative $F_C$ with radii can be explained by spatial heterogeneity over the urban landscape, and the mismatch between the modelling domain and flux source area. Regardless of the modelling domain size, traffic is the dominant $CO_2$ source, contributing 63–73% to the total $CO_2$ emissions, followed by metabolism (14–18%), respiration released by vegetation and soil (6–11%), and heating (6–9%). Vegetation photosynthesis offsets only 4–8% of the $CO_2$ emissions.

We highlight the importance in choosing more site-specific LAI parameters or $g_{max}$ when using SUEWS in modelling heat fluxes, before more advanced application regarding urban climate and hydrological modelling. Observations are needed to support more accurate parameterizations of biogenic $CO_2$ fluxes. We believe the bottom-up approach to model $F_C$ by SUEWS can be a promising tool in quantifying urban $CO_2$ emissions at the local scale, and therefore apply to capture the $CO_2$ emission hot spot, quantifying relative contribution of $CO_2$ sources, and assist to mitigate urban $CO_2$ emissions.

**Appendix A: Workflow of LAI parameters optimization**

In SUEWS, LAI influences the surface conductance, and subsequently $Q_E$ and $F_{pho}$ (Sect. 2). A workflow for parameter derivation for the LAI sub-model based on remotely-sensed data is designed for natural ecosystems (Omidvar et al., 2022). However, vegetation in urban areas behaves differently from natural ecosystems (Zhang et al., 2022), and needs to be considered separately. Therefore, we propose a workflow to obtain the parameters for urban area based on remotely-sensed LAI and

Covariance matric adaptation evolution strategy (CMA-ES). This workflow can also be applied to natural ecosystems. The related data and codes are openly available (Zheng et al., 2022).

Covariance matric adaptation evolution strategy (CMA-ES) is one of the strategies for numerical optimization of non-convex problems. It is based on the principle of biological evolution. The evolution strategy takes a certain number of individuals (candidate solutions) in a stochastic way, selects individuals based on the fitness, and repeats this process for generations so

that a better or an optimal solution is obtained. Adaptation of the covariance matrix amounts to learning a second order model of the underlying objective function. Compared with classic optimization methods, CMA-ES requires neither derivatives nor an objective function; it only requires the ranking of candidate solutions. Besides, CMA-ES outranks many of other optimization algorithms, performing especially strong on "difficult functions" or larger dimensional search spaces (Hansen et al.).

Taking Beijing as an example, the LAI parameters are optimized as the following:

1. City-level LAI derivation. The NASA Moderate Resolution Imaging Spectroradiometer (MODIS) product MOD15A2H with 8-day and 500-m resolution is first derived. The spatial average of LAI within the 6$^{th}$ ring area of Beijing is extracted from the year 2008 to 2016, treated as the city-level of LAI.

2. Scaling city-level LAI to the tree-level. We note that the city-level LAI might be noticeably lower than the LAI at tree level due to the presence of non-vegetated surfaces in urban area. Nonetheless, the city-level LAI provides the signals



of vegetation phenology (e.g., leaf-out, peak growing season, leaf-fall). In order to give a more realistic estimate of LAI at the tree-level, the LAI city-level needs to be scaled. Therefore, $LAI_{max}$ and $LAI_{min}$ need to be given manually, preferably based on observational studies over local species. An observational study in Beijing shows that the maximum of LAI falls between 5 and 6 $m^2\ m^{-2}$ (Wang et al., 2021). Here, $LAI_{max} = 6$ and $LAI_{min} = 0.1\ m^2\ m^{-2}$.

3. Spikes removal. Spikes occur occasionally due to instrument problems, uncertainties of retrieval algorithm and cloud
contamination. Therefore, data points are identified as spikes if:

$$|LAI - LAI_5| > \frac{\sqrt{LAI + LAI_5}}{\beta}, \tag{A1}$$

where $LAI_5$ is the moving median of 5 consecutive data points, and $\beta$ is constraint factor (here $\beta = 2$). The spikes are removed, marked as "sample discarded"; the other sample are kept, marked as "sample retained" and treated as "observed LAI". In the case shown here, 7 data points are discarded, accounting for 1.7% of the total sample.

4. Interpolation. The observed LAI are linearly interpolated between values to obtain a daily time series, marked as "interpolated LAI".

5. Parameters derivation using CMA-ES. The time series of interpolated LAI and $T_{air}$ are subjected to CMA-ES to optimize the parameters. Using the LAI model and the parameters derived, LAI is calculated and marked as "predicted LAI".

LAI model incorporated with the optimized parameters successfully reproduces the seasonal dynamics of observed LAI,
capturing the timing of growth, peak growing season, and senescence (Fig. A1 a). The model performance is remarkably good, with high $R^2$ (0.94) and low RMSE (0.4 $m^2\ m^{-2}$) (Fig. A1 b).

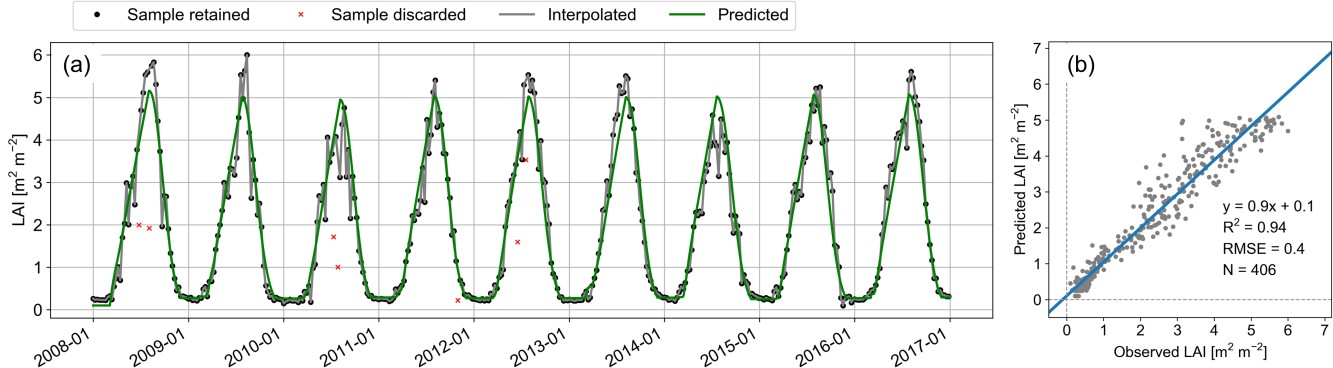

**Figure A1.** (a) Time series of leaf area index (LAI) sample retained, LAI sample discarded, interpolated LAI and predicted LAI, and (b) comparison of the predicted against observed LAI.



**Appendix B: Evaluation of WFDE5 reanalysis against observed meteorological variables**

To force SUEWS, local scale meteorological data within the inertial sublayer is required. However, they can be unavailable for the area and period desired. Reanalyses provide spatially and temporally complete datasets, which might make the modelling

run easier for users. Kokkonen et al. (2018) and Kokkonen et al. (2019) evaluated one of the reanalyses, WATCH Forcing Data ERA-Interim (WFDEI), suggesting that WFDEI can serve as the forcing of SUEWS, but should be corrected beforehand when the bias is large.

WFDE5 is a bias-corrected dataset of near-surface meteorological variables derived from the fifth generation of the European Centre for Medium-Range Weather Forecasts (ECMWF) atmospheric reanalysis (ERA5) (Cucchi et al., 2021). It is generated

using the same methodology as WFDEI, and provides a single layer at 0.5° spatial resolution and hourly temporal resolution. The evaluation of WFDE5 and the use of it as forcing data to SUEWS have been neglected so far. Here, we compare WFDE5 against observed meteorological variables including air temperature ($T_{air}$), relative humidity ($RH$), wind velocity ($U$) at 47 m, and incoming shortwave radiation ($K_{down}$) at 140 m on the IAP tower (Liu et al., 2012). The evaluation of $K_{down}$ is conducted from May 2010 to June 2011, and the rest from January 2010 to December 2011. All the observed variables are resampled

from 30-minute to 1-hour resolution.

**Table B1.** The height of WFDE5 and observed meteorological variables.

|  | WFDE5 | Observations |
|---|---|---|
| $T_{air}$ | 2 m | 47 m |
| $RH$ | near surface | 32 m |
| $U$ | 10 m | 47 m |
| $K_{down}$ | near surface | 140 m |

With the difference in height for each meteorological variable (Table B1), however, WFDE5 is close to the observed as a whole (Fig. B1). Compared with the observed, WFDE5 $T_{air}$ is lower, $RH$ higher, $U$ lower, and $K_{down}$ higher. WFDE5 may underestimate $T_{air}$, and overestimate $RH$, for neglecting the urban anthropogenic heat release; WFDE5 might overestimate $K_{down}$ due to insufficient consideration of aerosol's effect in decreasing solar radiation received by the urban surface. The

lower $U$ of WFDE5 can be explained by the lower height compared with the observations (Table B1). If the WFDE5 $U$ is adopted as forcing of SUEWS, aerodynamic resistance might be overestimated, and therefore $Q_E$ underestimated.

Admittedly, some meteorological variables of WFDE5 correlate poorly with the observations in a particular season (e.g., $R^2$=0.13 for $U$ in JJA). However, the overall high $R^2$, and low RMSE and MBE in magnitude suggest that WFDE5 provides reasonably good estimates of each meteorological variable (Table B2). Therefore, WFDE5 is adopted as the forcing data of

SUEWS in this study.

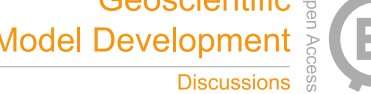

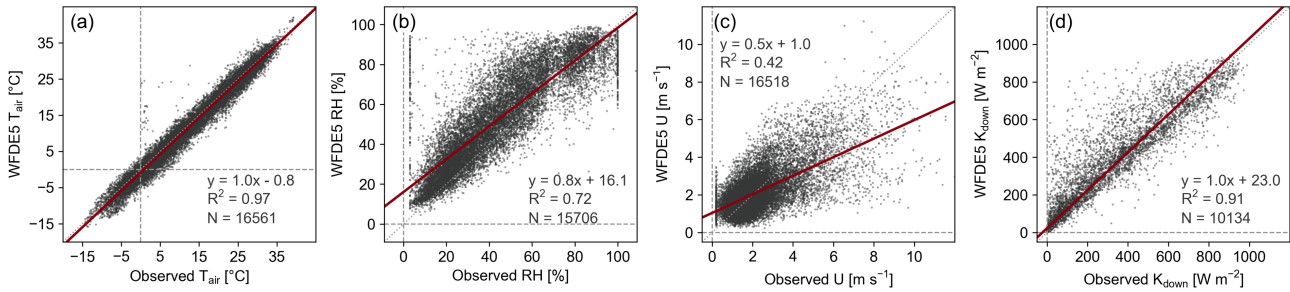

**Figure B1.** Comparison of WFDE5 and observed meteorological variables including (a) air temperature ($T_{air}$), (b) relative humidity ($RH$), (c) wind velocity ($U$) and (d) incoming solar radiation ($K_{down}$) at an hourly resolution.

**Table B2.** Statistics for WFDE5 compared against the observed meteorological variables.

|  | Season | average | | $R^2$ | RMSE | MBE | N |
| --- | --- | --- | --- | --- | --- | --- | --- |
|  |  | WFDE5 | Observed |  |  |  |  |
| $T_{air}$ (°C) | DJF | -2.1 | -1.2 | 0.83 | 2.3 | -1.0 | 4217 |
|  | MAM | 12.7 | 13.5 | 0.92 | 2.5 | -1.0 | 4100 |
|  | JJA | 26.2 | 26.8 | 0.77 | 2.1 | -0.5 | 3966 |
|  | SON | 13.8 | 14.2 | 0.94 | 1.9 | -0.4 | 4278 |
| $RH$ (%) | DJF | 41.8 | 32.4 | 0.77 | 13.4 | 9.3 | 4216 |
|  | MAM | 41.6 | 34.1 | 0.74 | 13.8 | 7.9 | 3986 |
|  | JJA | 66.6 | 59.9 | 0.71 | 12.6 | 5.1 | 3546 |
|  | SON | 60.2 | 49.3 | 0.50 | 20.6 | 10.9 | 3958 |
| $U$ (m s$^{-1}$) | DJF | 2.1 | 2.7 | 0.52 | 1.3 | -0.5 | 4217 |
|  | MAM | 2.7 | 2.8 | 0.45 | 1.4 | -0.1 | 4100 |
|  | JJA | 1.9 | 1.9 | 0.13 | 1.2 | 0.1 | 3949 |
|  | SON | 2.0 | 1.9 | 0.45 | 1.0 | 0.0 | 4252 |
| $K_{down}$ (W m$^{-2}$) | DJF | 117.0 | 99.2 | 0.94 | 50.6 | 19.5 | 2160 |
|  | MAM | 222.5 | 218.1 | 0.93 | 82.7 | 19.7 | 2941 |
|  | JJA | 216.4 | 189.3 | 0.86 | 110.5 | 38.2 | 2883 |
|  | SON | 140.9 | 124.7 | 0.93 | 58.9 | 16.5 | 2150 |





## Appendix C: Fitting maximum photosynthetic rate for vegetation type of grass/lawn

In order to find the maximum photosynthetic rate for the vegetation type of grass to be used in SUEWS simulation, the environmental response functions $g(T_{air})$, $g(\Delta q)$, $g(\Delta \theta)$, and $g(K_{down})$ in Eq. (14) were fitted to observations from an eddy covariance (EC) station (60°11'16.02"N, 24°49'56.85"E) situated in an urban lawn in Espoo, Finland. A 1.2 m high EC tower

was located at the centre of the urban lawn covering an area of 0.7 ha. The EC setup consisted of a three-dimensional sonic anemometer (Metek GmbH, Germany) for measuring the three wind components and sonic temperature, and a closed-path infrared gas analyser (LI-7200; LI-COR, Lincoln, NE, USA) for measuring the $CO_2$ and $H_2O$ mixing ratios. The gas analyser inlet was positioned 13 cm below the anemometer and air was drawn into the gas analyser using a 60 cm length of steel tube, having an inner diameter of 4.57 mm and a mean flow rate of 12 l min$^{-1}$. The tube was heated to avoid water vapor

condensation on tube walls. The raw EC data were sampled at 10 Hz and stored for post processing. The steps before 30-min flux calculations consisted of de-spiking, linear de-trending and planar fitting of the raw data.

The biogenic $CO_2$ flux $F_{c,bio}$ from EC flux measurements were partitioned into $F_{res}$ (obtained by fitting it exponentially to the measured $T_{air}$) and $F_{pho}$ (obtained by subtracting $F_{res}$ from $F_{c,bio}$). Then $F_{pho}$ was used as a dependent variable, whereas on-site measurements of net radiation (CNR4; Kipp&Zonen, Delft, Netherlands), air temperature and relative humid-

ity (HMP110 A15; Vaisala Oyj, Vantaa, Finland) and soil moisture (ML3 Thetaprobe; Delta-T, Cambridge, UK) were used to estimate the independent variables in $g(T_{air})$, $g(\Delta q)$, $g(\Delta \theta)$ and $g(K_{down})$. Atmospheric pressure from Finnish Meteorological Institute Kumpula station was used to calculate $\Delta q$. Additional reference values of soil properties (field capacity and wilting point), which were estimated to be same as in an urban lawn in Kumpula (Järvi et al., 2019), were used to calculate $\Delta \theta_{WP}$. Parameters $F_{pho,max}$ and $G_2 - G_6$ were fitted using a non-linear least-square approach.

Data from mid-July to end of August 2021 were used and in the fitting only data points with $K_{down} > 10$ W m$^{-2}$ and $\Delta q > 1$ g kg$^{-1}$ were selected (Havu et al., 2022a). After bootstrapping method to randomly select seven eighths of the data for 100 times, the final parameters fitted for grass were obtained as medians with the uncertainties as follows: $F_{pho,max} = 5.497 \pm 0.110$ $\mu$mol m$^{-2}$s$^{-1}$, $G_2 = 195.019 \pm 5.601$ W m$^{-2}$, $G_3 = 0.741 \pm 0.008$, $G_4 = 0.413 \pm 0.015$, $G_5 = 30.000 \pm 0.000$ °C, $G_6 = 0.500 \pm 0.000$ mm$^{-1}$. The value of $F_{pho,max}$ is used for grass/lawn in the Beijing simulation.





**Appendix D: Maximum conductance of urban green space in Beijing**

**Table D1.** Maximum conductance ($g_{max}$) for each vegetated surface.

|  | Deciduous tree | | | |
|---|---|---|---|---|
|  | *Sophora japonica* Linn. | *Populus tomentosa* Carr. | *Faxinus chinensis* Roxb. | *Ginkgo biloba* Linn. |
| Ratio[a] | 26.26% | 12.39% | 9.44% | 8.12% |
| $g_{max}$ (mm s$^{-1}$) | 9.0 | 5.2 | 6.1 | 4.1 |
| Reference | Xu et al. (2020) | Wang et al. (2018) | Xu et al. (2020) | Song et al. (2015) |
|  | Evergreen tree | Grass/lawn | | |
|  | *Pinus tabuliformis* | *Festuca arundinacea* Schreb. | *Poa pratensis* L. | *Zoysia japonica* Steud. |
| $g_{max}$ (mm s$^{-1}$) | 1.4[b] | 5.4 | 4.0 | 1.6 |
| Reference | Chen et al. (2021) | Wang et al. (2006) | Wang et al. (2006) | Wang et al. (2006) |

[a] obtained from a field survey over Beijing (Ma et al., 2019)

[b] obtained by dividing maximum canopy conductance by $G_1$ (=3.5)

The weighted average of $g_{max}$ is 7.0 mm s$^{-1}$ for deciduous tree weighted by population ratio of each species (Table D1). The average $g_{max}$ for grass is 3.7 mm s$^{-1}$. Note that when the species in the modelled area is known, we suggest the $g_{max}$ is selected accordingly. Here, we seek values that can represent the overall vegetation over the urban area in Beijing, and therefore the average is taken from different species.



**Appendix E: Comparison of parameters for modelling domain with different radii**

**Table E1.** Land surface fraction for modelling domain with different radii, where "Paved" denotes paved surface, "Bldgs" buildings, "EveTr" evergreen tree, "DecTr" deciduous tree, "Grass" grassland and lawn, "Bsoil" bare soil, "Water" water body.

| | Radius (m) | | | |
|---|---|---|---|---|
| | 500 | 750 | 1000 | 1500 |
| Fr_Paved | 0.39 | 0.32 | 0.46 | 0.46 |
| Fr_Bldgs | 0.18 | 0.21 | 0.24 | 0.24 |
| Fr_EveTr | 0.04 | 0.04 | 0.02 | 0.02 |
| Fr_DecTr | 0.18 | 0.18 | 0.11 | 0.01 |
| Fr_Grass | 0.20 | 0.24 | 0.16 | 0.17 |
| Fr_Bsoil | 0 | 0 | 0 | 0 |
| Fr_Water | 0.01 | 0.01 | 0.01 | 0.01 |

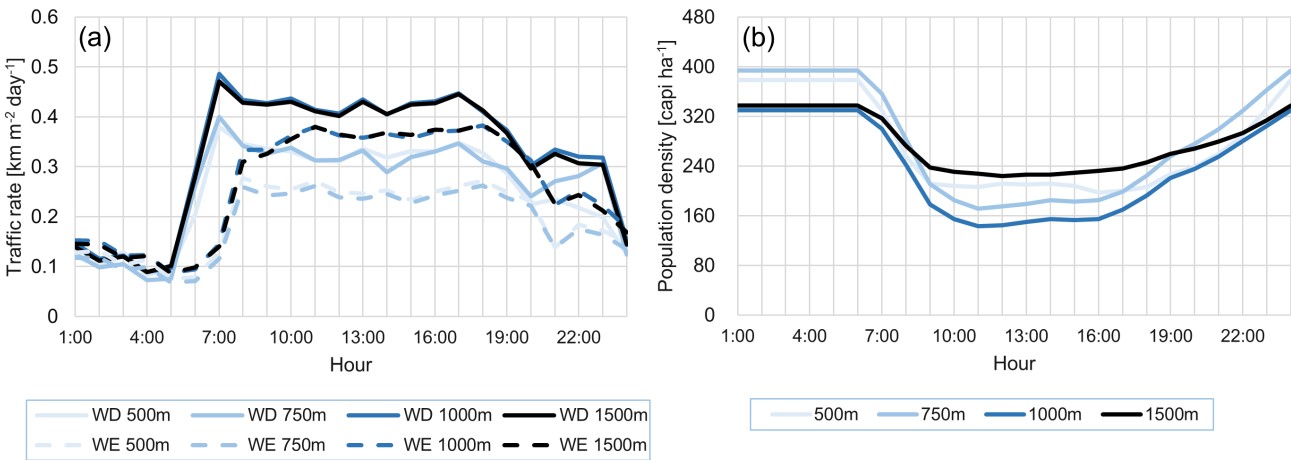

**Figure E1.** Diurnal cycles of (a) traffic rate and (b) population density for modelling domain with different radii, where WD denotes weekday and WE weekday.





*Code and data availability.* The data sets are openly available, including the complete model runs of SUEWS, the meteorological, radiation, and turbulent flux measurements, codes for LAI model optimization using CMA-ES, and codes to reproduce the statistics and figures (Zheng et al., 2022).

*Author contributions.* YZ, MH, HL and LJ conceptualized the study. YZ performed the SUEWS model runs, analyzed the results, and
prepared the figures. HL, XC, YW, and JA collected the data. HSL performed the fitting of maximum photosynthetic rate. LJ and HL supervised the study. All authors contributed to writing and preparing the manuscript.

*Competing interests.* Leena Järvi is a member of the editorial board of Geoscientific Model Development. The peer-review process was guided by an independent editor, and the authors have also no other competing interests to declare.

*Acknowledgements.* This study is funded by National Natural Science Foundation of China (Grant No. 42161144010), China Scholarship
Council (Grant No. 202104910363), Tiina and Antti Herlin Foundation (Grant No. 20200027), the Academy of Finland (Grant Nos. 321527 and 337549), and the Strategic Research Council established within the Academy of Finland (Grant No. 335201). We sincerely appreciate Mr. Weiyi Zuo from Institute of Acoustics, Chinese Academy of Sciences for his significant technical support in realizing CMA-ES method in this study.





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
