# Peer review of "Simulating heat and CO2 fluxes in Beijing using SUEWS V2020b: Sensitivity to vegetation phenology and maximum conductance"

_Geoscientific Model Development, 2022_

## Author Comment (AC1)

**Added tables in Supplementary Materials:**

**Table S4.** Adjusted albedo ($\alpha_i$) for different surfaces: buildings (Bldgs), paved surface (Paved), evergreen tree/shrub (Everg), deciduous tree/shrub (Dec), grass, and water following Ward et al (2016). The $\alpha_i$ for deciduous tree/shrub and grass are allowed to vary from a lower value in summer to a higher value in winter.

|  | Unit | Bldgs | Paved | Everg | Dec | Grass | Water |
|---|---|---|---|---|---|---|---|
| $\alpha_i$ | – | 0.12 | 0.10 | 0.10 | 0.12-0.18 | 0.18-0.21 | 0.10 |

**Table S5.** SUEWS model performance statistics with adjusted albedo ($\alpha_i$) (Table S4) for radiation fluxes, including incoming solar radiation ($K_{down}$), outgoing shortwave radiation ($K_{up}$), incoming longwave radiation ($L_{down}$), outgoing longwave radiation ($L_{up}$), and net radiation ($Q_N$) from May 2010 to June 2011.

|  | Season | $R^2$ | RMSE | MBE | N |
|---|---|---|---|---|---|
| $K_{down}$ | DJF | 0.94 | 52.3 | 19.5 | 2160 |
|  | MAM | 0.93 | 82.6 | 19.8 | 2940 |
|  | JJA | 0.86 | 110.4 | 39.3 | 2728 |
|  | SON | 0.93 | 59.6 | 16.6 | 2150 |
| $K_{up}$ | DJF | 0.88 | 8.3 | -2.5 | 2160 |
|  | MAM | 0.92 | 10.1 | 0.2 | 2940 |
|  | JJA | 0.88 | 13.6 | 4.6 | 2728 |
|  | SON | 0.91 | 8.4 | 0.6 | 2150 |
| $L_{down}$ | DJF | 0.74 | 16.0 | -2.4 | 2160 |
|  | MAM | 0.86 | 20.5 | -10.0 | 2940 |
|  | JJA | 0.68 | 18.1 | 8.8 | 2728 |
|  | SON | 0.88 | 23.7 | 12.9 | 2150 |
| $L_{up}$ | DJF | 0.80 | 15.7 | 4.0 | 2160 |
|  | MAM | 0.90 | 19.1 | 3.6 | 2940 |
|  | JJA | 0.79 | 22.5 | 10.7 | 2728 |
|  | SON | 0.90 | 18.9 | 9.4 | 2150 |
| $Q_N$ | DJF | 0.94 | 40.5 | 15.6 | 2160 |
|  | MAM | 0.93 | 64.6 | 6.0 | 2940 |
|  | JJA | 0.86 | 88.8 | 32.8 | 2728 |
|  | SON | 0.92 | 51.8 | 19.5 | 2150 |

Modified figures:

[Figure]

Modified **Figure 5**. Average diurnal cycle of input or modelled and observed hourly radiation fluxes by season, including (a–d) incoming solar radiation ($K_{down}$), (e–h) outgoing shortwave radiation ($K_{up}$), (i–l) incoming longwave radiation ($L_{down}$), (m–p) outgoing longwave radiation ($L_{up}$), and (q–t) net radiation ($Q_N$) from May 2010 to June 2011. Shaded area denotes the interquartile range. Note that only Kdown is input and the rest are model output.

[Figure]

Modified **Figure 6**. Annual and seasonal mean diurnal cycles of observed and modelled (a–e) latent heat flux ($Q_E$) and (f–j) sensible heat flux ($Q_H$) for the four model runs (case **base**, **gs**, **LAI**, and **gs_LAI**) in the year 2016. Shaded area denotes the interquartile range.

[Figure]

Modified **Figure 8**. Annual and seasonal average diurnal cycles of observed and modelled $CO_2$ flux ($F_C$) for the four model runs (case **base**, **gs**, **LAI**, and **gs_LAI**) in the year 2016. Shaded area denotes the interquartile range.

[Figure]

Modified **Figure 9**. Seasonal average diurnal cycles of modelled $CO_2$ flux ($F_C$) components by case **gs_LAI** in 2016. FcTraff denotes $F_C$ from on-road traffic, FcBuilding building, FcMetab human metabolism, FcRespi vegetation and soil respiration, and FcPhoto vegetation photosynthesis. Positive values indicate sources of $CO_2$ and negative values sinks with respect to the atmosphere. Shaded area denotes the interquartile range.

---

## Author Response (AR2)

Authors' note: our responses for the two iterations, i.e., Revised submission (page 2−35) and Correction (page 36−40) have been merged as shown below.

**Date: 11 Jun 2023, Iteration: Revised submission**

Dear Editor and Reviewers,

We appreciate all the valuable and helpful suggestions from the reviewers. The manuscript has been revised carefully. All the revisions and new references have been marked by tracked changes in the manuscript. The related codes and dataset have been also updated (Zheng et al., 2022). Please kindly note that 12 out of then total 15 figures have been revised. In the revised manuscript with tracked changes, only the new figures were demonstrated for readability. The editor's and reviewer's comments are in italics and our response is plain text. Our responses are shown below.

**Response to Topic editor's comment (11 Jun 2023)**

*I evaluate that the manuscript is within the scope of the journal and it meets an essential scientific quality.*
*When authors revise the manuscript with comments from reviewers, please add explanations for shading areas of figures 6, 8, and 9.*

We thank the editor for noticing the missing of explanations. The explanations for the shading areas were added accordingly. Please see the tracked changes in the manuscript.

**Response to Anonymous Referee #1 (11 Jun 2023)**

We appreciate all the valuable and helpful comments and suggestions from the reviewer. Those allowed us to improve the manuscript. The contents mentioned by the reviewer have been inspected or revised point by point accordingly. Our responses are as follows:

*This manuscript evaluates the simulation results of the SUEWS on radiation flux, turbulent heat flux and CO2 flux at a densely built neighbourhood in Beijing. Using the site-specific gmax and optimized LAI parameters, the modelling of turbulent heat fluxes is improved.*

*The Fc module of SUEWS is applied in Beijing for the first time, and the simulation results of CO2 flux are satisfactory, which makes it possible to quantitatively evaluate the contribution of various CO2 sources and sinks and facilitate comparison with other observation sites.*

*However, there are some problems in the analysis and discussion results. For example, the radiation parameterization scheme NARP does not involve parameters gmax and LAI, so there is no way to say "gmax and LAI parameters has only a minor impact on the modelled radiation fluxes".*

*Some of the analysis and discussion in the manuscript are relatively simple and one-sided. Please see the following specific comments.*

We thank the reviewer for highlighting the relevance of the manuscript and we agree that there is room to improve the analysis and discussion to make the expressions more accurate and scientifically sound. We agree that the expression of *"gmax and LAI parameters has only a minor impact on the modelled radiation fluxes"* needs to be rephrased. Besides, we conducted one additional model experiment and modified four of the figures. Below, we discuss how the analysis and discussion would be improved in detail.

*Line 167-176, Is the data during precipitation included in the deleted data?*

The data during precipitation have been discarded (Cheng et al., 2018) prior to flux data quality control steps mentioned in our manuscript. In order to make it clearer, the following paragraph has been modified.

Modified text (Line 167):

"The 30-min turbulent flux calculation procedures and quality controls were described in detail by Cheng et al (2018). Quality controls such as out-of-limit value removal, spike removal and dropout test were conducted on the 10 Hz data during the flux calculation. In order to exclude low-quality data caused by precipitation, dust, or other contamination on the sensor, the records with automatic gain control value $\geqslant$ 62 were discarded. On top of the procedures by Cheng et al (2018), the following quality control steps are performed …"

*In Table 2, is the TL set at -10 ℃ applicable to IAP, and is there any possibility that the air temperature in Beijing will be lower than -10 ℃?*

TL means at this temperature (-10 ℃ in this context) or below, the vegetation becomes totally dormant, the conductance becomes zero, and thus the vegetation evapotranspiration and photosynthesis switch off completely. This value was proposed to be applicable across a range of sites and conditions (Ward et al, 2016), and it has been adopted and validated by Järvi et al (2019) and Havu et al (2022). It is possible that air temperature in Beijing being lower than -10 ℃ in winter months, but the value TL is not dependent on the climate in Beijing. Therefore, TL= -10 ℃ is applicable to IAP area.

*Line 290-291, Generally, air temperature (including daily minimum air temperature, daily range of air temperature, accumulated temperature, etc.) is the main factor affecting urban vegetation phenology in Beijing. Vegetation in cities is irrigated frequently, so there are few cases where vegetation growth is limited by soil moisture, which is different from the non-urban areas Omidvar et al. (2022) studied. The optimization of the LAI model is of course necessary and recommended, but it is not reasonable to explain the factors affecting vegetation growth and senescence rate here.*

We thank the referee for the insight on the vegetation phenology controlling factors. We agree that air temperature is the main factor controlling urban vegetation phenology in Beijing. As the role of soil moisture was not well justified in the original text, we rephrased Line 290 to Line 291 as follows:

"In Beijing, the rainy season lasts from May to October, while the other time of the year is dry season (Liu et al., 2012). It is possible that the distinct dry season leads to a lack of soil moisture in spring and autumn and thus influences the LAI seasonal dynamics if there is no external water input (Omidvar et al., 2022), but the urban green spaces in Beijing are usually sufficiently or even excessively irrigated (Zhang et al., 2017). Observations also provided evidence to support the relationship between air temperature and phenological dynamics in the urban environment in Beijing (Lu et al., 2006; Luo et al., 2007). Therefore, the air temperature-dependent LAI model is applicable in Beijing, but the 'default' LAI parameters might be not suitable. We

recommend evaluating the LAI model when SUEWS is applied to a different city, and deriving the optimal LAI parameters if necessary."

*Line 295-299, the radiation parameterization scheme (NARP) in the SUEWS does not involve gmax and LAI, so the simulation results of radiation flux among 4 cases should be identical, and the R2 is 1, but why are the RMSE and MBE are not equal to 0?*

Previously, we found that $R^2$ was 1, while RMSE and MBE were negligible yet higher than 0, so we reported them as how they were. We did compare the output radiation flux components manually and found them, indeed, identical among the 4 cases. The values of RMSE and MBE were very likely to be introduced by error of Python numerical calculation (specifically, the round-off error in floating point numbers) during the calculation of the RMSE and MBE instead of NARP itself. We apologize for bringing unnecessary confusion. Note that this section has been moved to the Appendix upon to the suggestion from the second reviewer.

Modified text (Line 295−299):

"The radiation parameterization scheme Net All-wave Radiation Parameterization (NARP) does not involve $g_{max}$ or LAI. As expected, the four experiments (Sect. 4.4.1) give identical radiation flux components in the output. Therefore, only the case **gs_LAI** is further analyzed here."

*Line 307-308, In addition to the lower albedo of vegetation in summer, wet surface caused by frequent rainfall and radiation trapping caused by street canyons also lead to the decrease in surface albedo (Ao et al., 2016; Oke et al., 2017; Dou et al., 2019). Given the vegetated fraction is low, only adjusting the albedo for vegetation has a limited effect on improving the simulation results of radiation fluxes as stated by the author. Therefore, it is recommended to adjust albedos for all surface types in SUEWS, as Ward et al., (2016) did, in order to improve simulation results, especially Kup.*

We thank the referee for this constructive suggestion. Observational studies show that surface albedo reacts to surface wetness divergently: the surface albedo might decrease (Ao et al., 2016) or increase slightly (Dou et al., 2018) after precipitation, and the different behaviors were likely caused by the difference in surface geometric structure and materials. Under the current parameterization in SUEWS, the surface albedo is not related to surface wetness or street canyon trapping effect. The influence of these two factors might not be urgently introduced to the albedo parameterization to maintain the virtue of simplicity of SUEWS.

We conducted a new model run to support the idea that adjusting albedos for all surface types especially for the non-vegetative surfaces can help to improve the simulation of $K_{up}$. The new parameters and model performance statistics were demonstrated in Table S5 and Table S6, respectively. These two tables would be added to the Supplementary Materials rather than main text, because (1) the original model run had provided reasonably good performance on radiation components, especially on net radiation flux ($Q_N$), (2) adjusting albedos only had a minor impact on $Q_N$, giving a marginal increase in RMSE by $0-2.5$ W m$^{-2}$ (Table S6), and (3) adding these new contents to Supplementary Materials helps to avoid obscuring the main topic of this manuscript.

Modified text (Line 305 to Line 309):

"The annual bulk albedo for the modelling domain given to SUEWS is 0.14, which is relatively high but still consistent with the observations. Larger positive bias in $K_{up}$ is observed in summer than in winter. Surface albedo is influenced by many factors such as surface wetness and street canyon trapping effect (Ao et al., 2016; Dou et al., 2018), which have not yet been considered by SUEWS. By simply (1) adjusting the albedos for surface types following Ward et al. (2016), and (2) allowing albedo for vegetation vary from a lower value in summer to a higher value in winter (Table S5), the RMSE for $K_{up}$ decreases for all seasons, especially in summer (from 18.0 to 13.6 W m$^{-2}$), but this has only a minor impact on $Q_N$ modelling (Table S6)."

*Line 313, the overestimated Lup might be induced by the lower emissivity of the building materials but does the Kdown of the reanalysis dataset WFDES also play a role? After all, it can be seen from Figure 5a-d that Kdown is obviously overestimated, especially in summer.*

We agree.

Added text (Line 314):

"$L_{up}$ is also dependent on $K_{down}$ in NARP. Therefore, the overestimation of $L_{up}$ can be partly explained by the overestimated $K_{down}$ provided by WFDE5, especially around noon and in summer."

*Line 349-351, the Parameterization scheme of Fc does not include the parameter gmax but is related to LAI. At IAP, compared with anthropogenic emissions, the amount of CO2 absorbed by plant photosynthesis is relatively small, so the Fc is not sensitive to the improvement of LAI model. However, this is different from the case where QE simulation results are highly dependent on gmax and LAI. It is inappropriate to simply say that Fc is less sensitive to the improvement of gmax and LAI than QE without further explanation.*

We agree that it is necessary to clarify how the adjustments (LAI and $g_{max}$) affect the modelled respiration and photosynthesis.

Modified text (Lines 351−353):

"Under the current parameterizations, $F_{res}$ considers only air temperature (Eq. 15). The adjustments of $g_{max}$ and LAI parameters affect the modelled heat fluxes, influencing 2 m air temperature, and finally $F_{res}$, but the difference in annual $CO_2$ release from respiration is less than 0.01 kg C m$^{-2}$ yr$^{-1}$ among cases. The photosynthetic uptake is sensitive to the adjustments of $g_{max}$ and LAI parameters. In case **base**, the large values of $g_{max}$ allow too high evapotranspiration and $Q_E$. As a result, the average $\Delta\theta$ during January and June is larger than 105 mm, which is only 27 mm lower than the wilting point deficit ($\Delta\theta_{WP}$). Dryer soil furthermore lowers the surface conductance and photosynthetic $CO_2$ uptake through the limiting function of $g(\Delta\theta)$ (Eq. 8). As the local $g_{max}$ is introduced, soil remains moister with $\Delta\theta$ lower than 75 mm throughout the year, allowing a more favorable condition for the photosynthetic $CO_2$ assimilation. The $CO_2$ assimilated through photosynthesis is 0.57 kg C m$^{-2}$ yr$^{-1}$ in case **gs**, which is 0.21 kg C m$^{-2}$ yr$^{-1}$ higher than in case **base.** The LAI reduction in spring and autumn in case **gs_LAI,** on the other hand, directly limits surface conductance and photosynthesis (Eq. 14), leading to a decrease by 0.07 kg C m$^{-2}$ yr$^{-1}$ in annual photosynthetic $CO_2$ uptake when compared to case **gs**.

In SUEWS, photosynthetic and respiration rates are proportional to fractions of vegetated surfaces, which account for only 29% of the modelled area. The magnitude of $F_{pho}$ is substantially lower than the traffic emission, making the effect of photosynthesis, as well as its response to the adjustments of $g_{max}$ and LAI parameters, hardly visible in the $F_C$ diurnal cycle."

*Line 432, the NARP does not include the parameters gmax and LAI at all. They are not involved in the calculation of radiation fluxes, so it is not appropriate to say "hardly affected by gmax and LAI".*

We agree. We finally decided to remove this sentence from the conclusions.

*Line 433-435, For Beijing, plant phenology (leaf expansion time, leaf growth period, defoliation time, etc.) is generally more affected by temperature, not the transformation of dry-wet seasons.*

We agree. The content regarding soil moisture has been deleted.

*Line 438, Case gmax_LAI improved the simulation effect of QE, but I am not sure whether R2 increased by 0.02 can be called remarkable better.*

The performance on $Q_E$ modelling is better in terms of the decrease at the RMSE by 27.0 W m$^{-2}$ when compared to case **base**. The word "remarkably" was replaced with "noticeably".

Modified text (Line 438):

"By incorporating the local LAI parameters and $g_{max}$, SUEWS simulated the heat fluxes noticeably better, increasing $R^2$ by 0.03 (0.30) and decreasing RMSE by 27.0 (23.7) W m$^{-2}$ for $Q_E$ ($Q_H$), and showing more realistic seasonal dynamics when compared to EC observations."

Technical corrections/suggestions/language edits (not exhaustive!)

==================

*Line 90-91, "while outgoing longwave radiation (Lup) is estimated by a surface emissivity, α, Kdown, Lup and Tair". The second Lup should be Ldown.*

Sorry for the mistake. The mistake has been corrected.

*In Figure 5, the shaded area is recommended to be represented by the IQR rather than the standard deviation to display more data information. The same cases are in Figures 6, 8, and 9.*

We changed all the shaded areas shown in this manuscript from standard deviation to the IQR in order to better display the data statistical distribution. Please find the updated figures in the revised manuscript.

*Line 309, that the average seasonal and diurnal cycles of Ldown are well captured by the model are shown in Fig.5 i-l rather than Fig.4 i-l.*

The text has been modified accordingly.

*Line 311, the full name of the NARP (Net All-wave Radiation Parameterization scheme) should be given when it is mentioned for the first time in the submitted manuscript.*

The full name of NARP has been given accordingly.

*Line 469, The cited reference is short of publication year.*

The missed information has been added.

*In lines 33, Line 43, Line 360, Line 382-383, and Line 421, the arrangement of references is not consistent.*

Thank you for noticing the inconsistency. They are now placed in alphabetical order.
* * *
**References**

Dou, J. X., Grimmond, C. S. B., Cheng, Z. G., Miao, S. G., Feng, D. Y., and Liao, M. S.: Summertime surface energy balance fluxes at two Beijing sites, Int. J. Climatol., 39, 2793–2810, doi:10.1002/joc.5989, 2019.

Oke, T. R., Mills, G., Christen, A., Voogt, J. A. (2017) Urban Climates. Cambridge: Cambridge University Press. 134-137pp.    https://doi.org/10.1017/9781139016476

Ao X, Grimmond C S B, Liu D, et al. Radiation fluxes in a business district of Shanghai, China[J]. Journal of Applied Meteorology and Climatology, 2016, 55(11): 2451-2468.

Cheng X L, Liu X M, Liu Y J, et al. Characteristics of $CO_2$ concentration and flux in the Beijing urban area[J]. Journal of Geophysical Research: Atmospheres, 2018, 123(3): 1785-1801.

Lu P, Yu Q, Liu J, et al. Advance of tree-flowering dates in response to urban climate change[J]. Agricultural and Forest Meteorology, 2006, 138(1-4): 120-131.

Luo Z, Sun O J, Ge Q, et al. Phenological responses of plants to climate change in an urban environment[J]. Ecological Research, 2007, 22: 507-514.

Zhang X, Mi F, Lu N, et al. Green space water use and its impact on water resources in the capital region of China[J]. Physics and Chemistry of the Earth, Parts A/B/C, 2017, 101: 185-194.

Dou J, Grimmond S, Cheng Z, et al. Summertime surface energy balance fluxes at two Beijing sites[J]. International Journal of Climatology, 2019, 39(5): 2793-2810.

Omidvar H, Sun T, Grimmond S, et al. Surface Urban Energy and Water Balance Scheme (v2020a) in vegetated areas: parameter derivation and performance evaluation using FLUXNET2015 dataset[J]. Geoscientific Model Development, 2022, 15(7): 3041-3078.

Liu H, Feng J, Järvi L, et al. Four-year (2006–2009) eddy covariance measurements of $CO_2$ flux over an urban area in Beijing[J]. Atmospheric Chemistry and Physics, 2012, 12(17): 7881-7892.

Ward H C, Kotthaus S, Järvi L, et al. Surface Urban Energy and Water Balance Scheme (SUEWS): development and evaluation at two UK sites[J]. Urban Climate, 2016, 18: 1-32.

Järvi L, Havu M, Ward H C, et al. Spatial modeling of local-scale biogenic and anthropogenic carbon dioxide emissions in Helsinki[J]. Journal of Geophysical Research: Atmospheres, 2019, 124(15): 8363-8384.

Havu M, Kulmala L, Kolari P, et al. Carbon sequestration potential of street tree plantings in Helsinki[J]. Biogeosciences, 2022, 19(8): 2121-2143.

**Response to Anonymous Referee #2 (11 Jun 2023)**

We greatly appreciate all comments from the reviewer. The suggestions are helpful and some of them are constructive. These comments and suggestions enabled us to improve our manuscript, in terms of both the structure and content. We have carefully addressed the concerns of the reviewer and have made a revision of the manuscript. The reviewer's comments are in italics and our response is plain text. Please find the detailed responses below.
* * *
*This study applies the urban land surface model SUEWS on a neighborhood in Beijing to simulate the energy and CO2 exchange dynamics. A meteorological tower is located in the center of the study area, which provides observations of turbulent heat/CO2 fluxes and the four radiation components. The flux observations are used as reference to evaluate the model simulations and investigate the performance of the model under different parameterizations of the vegetated surfaces, focusing on vegetation phenology and conductance. The study concludes that is very important to adjust the vegetated surface parameterization according to site-specific vegetation information, especially for the accurate estimation of the turbulent heat fluxes.*

*This study contributes to the literature with insights on how urban vegetation affects the energy balance and the CO2 fluxes at local scale. Such information is still scarce in the literature, especially regarding the CO2 fluxes, and can help improve future model parameterizations and wider applications of urban land surface modeling for climate change mitigation and urban resilience planning.*

*There are however some problematic and unclear parts in the paper that deserve more attention (see general concerns below). Furthermore, even though the manuscript is in general well-structured and easy to follow, phrasing and grammar can be improved throughout the text. Some examples are given in the specific comments below.*

We thank the reviewer for highlighting the novelty of the manuscript. Based on the comments, we made some major revisions to the manuscript including: (1) revisiting the LAI optimization method, i.e., the CMA-ES, with a remotely-sensed LAI time series at a higher spatial resolution, (2) carrying out new model experiments to re-evaluate the model performance on turbulent fluxes with the updated LAI parameters, and (3) moving one result section (Section 5.2 Evaluation of radiation fluxes) to the Appendix. The major conclusions are basically consistent with the previous ones. A new version of the related data and codes has been uploaded (Zheng et al., 2022). Below, we discuss how the reviewer's suggestions have helped to improve the manuscript in detail.

*General concerns:*

*1. I am very skeptical regarding the LAI model optimization method. It is very surprising to see that the Authors have used the MODIS LAI/FAPAR 500 m resolution product which is based on a sophisticated 3D radiative transfer approach to derive LAI. If I am not mistaken, such approach would only be applicable on specific biomes and not on urban areas. The resolution of MODIS is too low to discern the green areas within an urbanized landscape. I would expect that such product would have omitted or at least flagged the areas that are not within its biome specifications. I recommend the Authors to double check the product and its quality flags. It is very probable that the LAI estimations of this product over Beijing are very unreliable.*

*Moreover, even if the LAI product was reliable, it is anyway challenging to assume that the phenology patterns derived by such a big area (ca. 40 km x 40 km, 6th ring area) would be representative of your case study (ca. 1 km2). The vegetation types and the management practices would be very diverse across such a huge area.*

*I suggest that the Authors would revisit their LAI optimization method by using high resolution satellite datasets, such as Sentinel-2 or Landsat, or field observations (e.g. phenocam imagery or field measurements).*

We thank the reviewer for the very constructive insight regarding how to select the input data for the LAI model optimization. We agree with the reviewer and have updated this part. Before the new LAI optimization, we checked the remotely-sensed LAI (MODIS LAI/FAPAR 500 m resolution product) over an urban park, the Beijing Olympic Forest Park (OFP). This park is the largest urban park in Asia with an area of 690 ha and a vegetation coverage of 90% (Chen et al., 2013). OFP undergoes common urban green space management practices, such as mowing and irrigation (Zhang et al., 2015). The LAI, on the one hand, showed a noticeably slower greening phase compared to the original "default" SUEWS pattern that reached the maximum LAI on early April. On the other hand, it showed a more rapid greening phase than the previous optimized LAI time series (Fig. R1). Therefore, we agree it is necessary to re-calculate the parameters describing the development of LAI.

In the new manuscript version, we used Landsat 7 data to calculate LAI of the vegetation in an adjacent park near the measurement tower (i.e., the IAP tower). With the new LAI time series, we revisited the optimization method. The related text in the main text and Appendix A, figures, and tables were updated accordingly.

[Figure]

Figure R1. Mean seasonal cycle of leaf area index (LAI) over the Olympic Forest Park in Beijing for the years 2011-2020. The time series were derived from the MODIS LAI/FAPAR 500 m resolution product (MCD15A2H). The shaded area denotes the standard deviation.

Modified text (Lines 177−181):

"To optimize the behavior of LAI, a six-year time series (2011–2016) of LAI over an adjacent park near the IAP tower is calculated from the atmospherically corrected surface reflectance provided by USGS Landsat 7 Enhanced Thematic Mapper + (ETM+) (30 m spatial resolution) via the Google Earth Engine Data Catalog (Masek et al., 2006). The atmospherically corrected surface reflectance bands have been preprocessed using the scaling factors from the metadata. Next, the enhanced vegetation index (EVI) is calculated using the formula (Huete et al., 1997),

$$EVI = 2.5 \times (NIR - RED)/(\text{NIR} + 6 \times \text{RED} - 7.5 \times \text{BLUE} + 1),$$

where NIR, RED and BLUE are the near-infrared, red and blue bands, respectively. The EVI is further used to calculate LAI with the formula (Boegh, 2002),

$$LAI = 3.618 \times \text{EVI} - 0.118.$$

The LAI and air temperature time series are subjected to optimization using Covariance matric adaptation evolution strategy (CMA-ES) (Appendix A). Before the optimization process, values larger than 10 $m^2$ $m^{-2}$ and negative values are considered outliers and removed; values during December and January are set to a fixed value, i.e., the average of these months (0.2 $m^2$ $m^{-2}$), to reduce the noise in winter and improve the optimization performance. More details can be found at Appendix A. The related data and codes are openly available to reproduce the results (Zheng et al., 2022)."

Since the input data have been replaced with the Landsat 7 LAI, the process of the optimization was also slightly changed.

Modified text (Lines 470−484):

"1. LAI derivation. A six-year time series (2011–2016) of LAI of the vegetation in an adjacent park near the IAP tower is calculated from the atmospherically corrected surface reflectance provided by USGS Landsat 7 Enhanced Thematic Mapper + (ETM+) (30 m spatial resolution) via the Google Earth Engine Data Catalog (Masek et al., 2006). The time series is treated as the "original LAI".

2. Spikes removal. There are outliers in the LAI time series caused by instrument problems, uncertainties of retrieval algorithm, and cloud contamination. Values larger than 10 $m^2$ $m^{-2}$ and negative values are first removed. The LAI values during December and January are set to a fixed value, i.e., the average of these months (0.2 $m^2$ $m^{-2}$), in order to reduce the noise in winter and improve the optimization performance."

3. Scaling the original LAI to the canopy level. The original LAI might be noticeably lower than the measured LAI at the canopy level over a homogeneous vegetated surface. Nonetheless, the original LAI provides the signals of vegetation phenology (e.g., leaf-out, peak growing season, leaf-fall). In order to give a more realistic estimate of LAI at the canopy level, the original LAI needs to be scaled. …. Here, the original LAI is scaled to allow the optimized LAI to reach 5−6 $m^2$ $m^{-2}$ in the peak growing season as reported by an observational study in Beijing (Wang et al., 2021). The canopy-level LAI is marked as the "input LAI" for the process of optimization and marked as the "observed LAI" for the process of evaluation."

Modified text (Lines 490−491):

"The input Landsat 7 LAI fluctuates greatly in summer, but the CMA-ES method provides a good estimate of the LAI seasonal dynamics, indicating that the CMA-ES is a useful tool that can handle input data contaminated by noise (Fig. A1 a). The model performance is overall good (with $R^2$ = 0.74 and RMSE = 1.2 $m^2$ $m^{-2}$) (Fig. A1 b)."

Admittedly, the LAI simulated with the parameters from Landsat 7 does differ from the previous one from MODIS. However, the optimized curve also indicates a later peak and an earlier decrease of LAI than the control run (case **base**).

Modified text (Lines 278−284):

"The control case **base** simulates the onset of leaf growth and the ending of senescence reasonably well (Fig. 3). The performance of LAI modelling is further improved after the optimization (Appendix A). In the case **base**, modelled LAI starts

to increase rapidly from day of year (DOY) 70 and plateaus at DOY 105, which is too early when compared to the remotely sensed LAI (Landsat 7 LAI). Optimized LAI starts to grow at the same time but slightly slower and peaks 20 days later than case **base**. In autumn, LAI modelled by case **base** drops rapidly at DOY 310, while optimized LAI starts to decline rapidly at DOY 267. LAI model with optimized parameters is better at capturing the behavior of senescence than in the control case **base**."

The SUEWS model was rerun, the model performance statistics were re-calculated, and the related figures were remade. The updated LAI parameters changed all the later figures, one of the tables, and some of the numbers and statements in the main text (shown as track changes in the manuscript).

The major conclusions of this paper are basically consistent with those in the previous manuscript: the model performance on turbulent heat fluxes by the case **gs_LAI** (with both $g_{max}$ and LAI parameters adjusted) is still the best among all the cases, but we state that the role of the $g_{max}$ adjustment is more important than the LAI adjustment.

*2. The part of the paper that presents the model performance regarding the CO2 fluxes is not sufficiently developed. There is a lack of clarity on how the two model adjustments (LAI, gmax) affect the modelled photosynthesis and respiration. There are different ways that such parameters would affect the vegetation and soil processes. In the manuscript is seems that the two parameters have opposite effects on photosynthetic performance. LAI reduction means Fpho reduction, gmax reduction probably induces higher Fpho due to larger soil water content. However, higher soil water content would also induce higher soil respiration, but I understand that this last effect is not included in SUEWS.*

We agree that it is necessary to clarify how the adjustments (LAI and $g_{max}$) affect the modelled respiration and photosynthesis.

Modified text (Lines 351−353):

"Under the current parameterizations, $F_{res}$ considers only air temperature (Eq. 15). The adjustments of $g_{max}$ and LAI parameters affect the modelled heat fluxes, influencing 2 m air temperature, and finally $F_{res}$, but the difference in annual $CO_2$ release from respiration is less than 0.01 kg C m$^{-2}$ yr$^{-1}$ among cases. The photosynthetic uptake is sensitive to the adjustments of $g_{max}$ and LAI parameters. In case **base**, the large values of $g_{max}$ allow relatively large evapotranspiration (namely $Q_E$). As a result, the average $\Delta\theta$ during January and June is larger than 105 mm, which is only 27 mm lower than the wilting point deficit ($\Delta\theta_{WP}$). The dry soil lowers the surface conductance and photosynthetic $CO_2$ uptake through the limiting function of $g(\Delta\theta)$ (Eq. 8). As the local $g_{max}$ is introduced, the soil remains moister with $\Delta\theta$

lower than 75 mm throughout the year, allowing a more favorable condition for the photosynthetic $CO_2$ assimilation. The $CO_2$ assimilated through photosynthesis is 0.57 kg C m$^{-2}$ yr$^{-1}$ in case **gs**, which is 0.21 kg C m$^{-2}$ yr$^{-1}$ higher than in case **base.** The LAI reduction in spring and autumn in case **gs_LAI,** on the other hand, directly limits surface conductance and photosynthesis (Eq. 14), leading to a decrease by 0.07 kg C m$^{-2}$ yr$^{-1}$ in annual photosynthetic $CO_2$ uptake when compared to case **gs**.

In SUEWS, photosynthetic and respiration rates are proportional to fractions of vegetated surfaces, which account for only 29% of the modelled area. The magnitude of $F_{pho}$ is substantially lower than the traffic emission, making the effect of photosynthesis, as well as its response to the adjustments of $g_{max}$ and LAI parameters, hardly visible in the $F_C$ diurnal cycles."

*More importantly, the modelled diurnal Fc patterns are not matching the observations during summer and to a lesser extent during spring and autumn months. The observed Fc patterns show clear seasonal changes. Morning Fc is decreasing during summer and increasing during winter, while the evening peak seems to be consistent during all seasons. This morning flux seasonal variability is not captured by the model. The Authors claim that the mismatch could be due to an underestimation of photosynthetic performance, but this is not supported by some evidence.*

*In order to gain a better understanding and interpretation of the results, I suggest that the Authors should do some further analyses: i. examine if the diurnal traffic patterns change seasonally in the study area, ii. examine if there are specific diurnal wind patterns for each season that would affect the observed land cover fractions per season and hour of day, iii. perform an analysis of the observed Fc according to wind sectors to investigate if there are "peculiar diurnal patterns" that would indicate the presence of point sources or wind sectors that are more affected by the green areas. iv. try to find if there are some unaccounted sources in the area (e.g. emissions from commercial/industrial buildings) from some emission inventory (if available).*

We thank the reviewer for providing very helpful suggestions to interpret the $F_C$ observations. We need to first point out that the model performance over the FC diurnal cycle is reasonably good as compared to a previous study (Järvi et al., 2019).

*i.  examine if the diurnal traffic patterns change seasonally in the study area*

The seasonal variation of diurnal traffic patterns is, unfortunately, unknown in the study area, and only the diurnal cycles of traffic rate on weekday/weekend are available. However, we would not assume that there is a marked seasonal variation in traffic rate. The traffic rates are expected to change somewhat during public holidays, most importantly during the spring festival holiday on late January and the National

Day holiday on early October lasting approximately one week. Summer holiday is usually only arranged for a small fraction of people, such as students and teachers. Therefore, it is unlikely that the public holidays have a marked influence in traffic rate on the seasonal scale.

ii. *examine if there are specific diurnal wind patterns for each season that would affect the observed land cover fractions per season and hour of day, iii. perform an analysis of the observed Fc according to wind sectors to investigate if there are "peculiar diurnal patterns" that would indicate the presence of point sources or wind sectors that are more affected by the green areas.*

This is a good point. We found that, indeed, the diurnal wind pattern varies with season, and the observed $F_C$ with wind direction noticeably (Fig. S2). However, we found that the vegetation fraction or road lane length could hardly serve as an indicator of $F_C$ in a particular season (Fig. S2). We modified the text in the manuscript to discuss the wind direction dependency.

Figure S2 was added to Supplementary materials.

Modified text (Line 365):

"There are multiple reasons to explain the difficulty in accurately capturing the diurnal cycle of the observed $F_C$ for each season. First, the influence of the underlying seasonal variation in the diurnal wind pattern is not considered. The observed $F_C$ varies noticeably with wind direction, and at the same time, the diurnal cycle of wind direction frequency varies with season. This makes the $F_C$ diurnal cycle from the NW quadrant more "seen" in winter and spring, while the diurnal cycle from SE more "seen" in summer and autumn (Fig. S2). SUEWS cannot consider this $F_C$'s wind-direction dependency as it simulates the overall flux from the simulation domain. Second, atmospheric stability influences the real-time footprint fetch of $F_C$ (Crawford and Christen, 2015), but this is not considered in our study. Third, there might be biases in simulating the seasonal cycles of $F_C$ component. It is possible that SUEWS underestimates the vegetation photosynthetic rate or overestimates the $CO_2$ release from respiration due to the lack of site-specific parameters. Nonetheless, the model performance over the $F_C$ diurnal cycle is reasonably good as compared to a previous study (Järvi et al., 2019)."

iv. *try to find if there are some unaccounted sources in the area (e.g. emissions from commercial/industrial buildings) from some emission inventory (if available)*

There is no industrial source of $CO_2$ in the study area to the best of our knowledge, but boiler plants for heating are common. We have investigated several boiler plants for space heating through interviews, but these are not treated as point sources; rather, they were calculated indirectly from the anthropogenic heat flux ($Q_F$) estimate and were included in the building $CO_2$ emissions (Eq. 13). The information about the boiler plants and the reason why we did not treat them as point sources have been added to the main text (see the response to the Lines 216 – 218 for details).

*3. A part of the paper focusses on the modelled radiation fluxes. However, it is not entirely clear how these are affected by the vegetation parameterization in SUEWS. Vegetation phenology and conductance would naturally affect radiation balance by modifying the surface albedo and emissivity over time but also by affecting the upwelling longwave radiation due to the cooling effect of evapotranspiration and the shading. To what extent are these processes directly or indirectly simulated by SUEWS? If they are not involved in the simulations, I wonder if the radiation fluxes evaluation is a relevant part of the manuscript. It is good to report the model performance, but if it is not connected to the study's main objectives, then it could be moved to an appendix or a supplementary file. Also, a discussion on the model shortcomings in respect to the vegetation effects on radiation fluxes would be relevant.*

We agree that radiation fluxes are less relevant to the main topic of $CO_2$ flux. These were presented because they (1) are needed to correctly calculate $Q_N$ and further the turbulent heat fluxes while $K_{down}$ is critical for photosynthetic uptake of $F_C$, and (2) the radiation flux observations were used to evaluate the SUEWS radiation flux parameterization in Beijing for the first time, which could provide valuable information to readers. Therefore, we placed this section in the main text while keeping it tight.

The vegetation is connected to the radiation flux parameterizations through surface albedo in SUEWS. We have conducted an additional model experiment to examine to what extent the model performance would be improved if the surface albedo for vegetative surfaces is allowed to vary with season in SUEWS. We found that model performance in $K_{up}$ increased (with a decrease in RMSE in summer by 4.4 W m$^{-2}$), but the performance in $Q_N$ was not improved (with a marginal increase in RMSE by $0-2.5$ W m$^{-2}$) (see also the authors' response, AC1: 'Reply on RC1' in the interactive discussion for more details). Considering SUEWS performed well in modelling radiation fluxes overall with the parameters from previous studies over other cities, we believed it is fine to neglect the seasonal variation of vegetative surface albedo in the study area. The vegetation emissivity evolves along with the vegetation seasonal

dynamics as well, but it might not be urgently introduced to the current radiation parameterization to maintain the virtue of simplicity of SUEWS.

In the current SUEWS parameterizations, radiation flux components are not connected to vegetation $g_{max}$ or LAI dynamics. To avoid obscuring the objectives of this manuscript, we decided to move the section to the Appendix.

Added text (Line 238):

"Model performance of radiation fluxes is evaluated prior to the simulation of turbulent heat fluxes. The results show that SUEWS is applicable to provide realistic estimates of radiation fluxes in the study area despite the absence of site-specific parameters (Appendix C)."

*Specific comments:*
*Line 2 and throughout the text: the term "sink" is used several times in the text to describe the negative CO2 flux. I believe this term cannot be used to describe the flux sign but to characterize the behavior of an ecosystem in the long term. The right term, as opposed to CO2 emissions, would be "CO2 uptake".*

The term "sink" in Line 2 was intended to describe the urban green space as an ecosystem. To avoid misuse, the term "$CO_2$ sink" was inspected, and all of them were rephrased according to the context. Changes were made to the original Line 2, 51, 127, 385 and 386.

*Lines 6 – 7 and throughout the text: "For **the** simulation of ....", "In **the** model evaluation, ....". In several places across the text the use of the grammatical article "the" is omitted. I suggest the Authors to have the text revised again for English phrasing and grammar.*

Thank you for noticing the mistakes. The grammatical article was added accordingly. The text has been revised for English phrasing and grammar again. Please see tracked changes for details.

*Line 24 - 25: sentences unclear, please rephrase.*

Modified Text (Line 22-25):

"Urban expansion has reshaped the morphological, thermal, and dynamical properties of the land surface (Grimmond and Oke, 2006; Oke, 1995; Zhu et al., 2016). In addition, intensive human activities in urban areas have caused a large quantity of greenhouse gas emissions (Marcotullio et al., 2013; Velasco and Roth, 2010). Both factors have influenced urban climate from micro to regional scales (Johansson and Emmanuel, 2006; Sarangiet al., 2018; Tan et al., 2010)."

*Lines 31 - 32: recheck the grammar in this sentence.*

Modified Text (Lines 31−32):

The results from the First International Urban Land Surface Model Comparison Project **suggested** that the most important processes for urban surface energy balance were radiative and vegetation processes (e.g., vegetation fraction, seasonal cycle of vegetation phenology) (Grimmond et al., 2010; Best and Grimmond, 2015; Nordbo et al., 2015).

*Line 59: "... imply **that** the sub-models ...".*

The phrase has been corrected accordingly.

*Line 64: In the main objectives the Authors state that they aim to evaluate the model under different vegetation parameterizations against radiation and turbulent fluxes. Is the radiation part of the model relevant? See main concern No. 3.*

The good performance of radiation modelling is the precondition of turbulent fluxes modelling. However, considering that the radiation modelling is not relevant to vegetation parameterizations (see the response to main concern No. 3), we rewrote the main objectives.

Modified Text (Lines 63−64):

"The main aims of this study are (1) to evaluate the model performance of SUEWS using different vegetation parameters (default and site-specific) against turbulent flux ($Q_E$, $Q_H$ and $F_C$) measurements, …"

*Lines 64 - 65: Using the term "partition Fc" in this sentence implies the use of a top-down approach. However, you do not apply any partitioning of the observed Fc in this study. The different Fc components are modelled separately by SUEWS (bottom-up). Overall, it is hard to assess the modeled contributions of each Fc component just by comparing to the measured net Fc.*

We agree that the term "partition $F_C$" was misused in this context. Modified text (Lines 64−65):

"The main aims of this study are (1) ..., and (2) to estimate the anthropogenic and biogenic components' contributions to the $F_C$ with the bottom-up modelling approach by SUEWS."

*Lines 126 – 127: More accurately: Fpho is the CO2 uptake by photosynthesis and Fres is the CO2 release by soil and vegetation respiration.*

Modified text (Lines 126−127):

"… $F_{pho}$ $CO_2$ uptake by photosynthesis, and $F_{res}$ $CO_2$ release by soil and vegetation respiration."

*Line 127, Eq. 10: The negative sign of Fpho is not indicated.*

Added text (Line 128):

"$F_{pho}$ has a negative sign while the rest of the $F_C$ components have a positive sign."

*Line 129: Repeated use of "based on", consider replacing once with "with".*

Modified Text (Lines 129−130):

"$F_M$ and $F_V$ are estimated with an inventory approach, i.e., based on population density or traffic rate, and their emission factors (EFs)."

*Lines 133 -134: The descriptions of the terms Ha,h,d and CM are not very clear.*

To make it clear, we split the $H_{a,h,d}$ into two terms ($PP_{h,d}$ and $AP_{h,d}$) and rewrote the descriptions.

Modified text (Lines 133−135):

"$F_{M,h,d} = p_{h,d} \cdot PP_{h,d} \cdot AP_{h,d} \cdot C_M,$

where $p_{h,d}$ is the daily average population density (cap ha$^{-1}$), PP,h,d population diurnal profile by hour, $AP_{h,d}$ activity level diurnal profile by hour, and $C_M$ $CO_2$ released per person ($\mu$mol $CO_2$ s$^{-1}$ cap$^{-1}$). These four parameters are given to SUEWS separately. The $p_{h,d}$ and $PP_{h,d}$ reconstruct the diurnal population density cycle. $AP_{h,d}$ scales the $C_M$ to vary between nighttime minimum and daytime maximum values ($C_{M(min,max)}$) to indicate the diurnal cycle of per capita human metabolic intensity."

The curves of $AP_{h,d}$ were also added to Fig. 2.

*Lines 133 – 146: The units used in the parameters within the anthropogenic emission models are very confusing and do not match in some cases between the text and Table 1. Consider describing the units and the conversions to µmol m-2 s-1 more carefully.*

The units demonstrated in the main text (i.e., kg km$^{-1}$ veh$^{-1}$) are the units required by SUEWS as external parameters. Emission factors are also usually reported in this unit which makes it easier for readers to compare. The unit conversions are done within the SUEWS model. To make it clearer, we added the parameter values with the conversed unit in parentheses to the Table 1 (Please see the tracked changes in the revised manuscript).

*Eq. 13: The term QF,cool is not explained.*

$Q_{F,cool}$ is the anthropogenic heat flux originating from cooling of the buildings. However, building cooling in Beijing is achieved through electrical devices (air conditioners) which do not involve on-site $CO_2$ emissions. $Q_{F,cool}$ was accidentally left to the equation but it was now removed. We apologize for the mistake.

*Lines 142 – 144: The descriptions of frheat and frnonheat are not complete and this causes confusion. As described by Järvi et al. (2019), these are the fractions of fossil fuels used for heating and other non-heating uses within the study area (i.e. local emissions). I would assume that frheat considers also the fuels used for cooling in the study area.*

As mentioned above, no $CO_2$ emissions are related to $Q_{F,cool}$ and thus it does not need to be considered in $fr_{heat}$. The terms $fr_{heat}$ and $fr_{nonheat}$ indicate how many percentages of the building heating and building energy use are caused by on-site fossil fuels used, respectively. In our study area, $fr_{heat}$ is related to the use of boilers for heating; $fr_{nonheat}$ is related to combustion from domestic cooking (the use of gas stove) by residents (see Section 4.3).

Modified text (Lines 143−146):

"…where $fr_{heat}$ is the fraction of fossil fuels used for heating, $Q_{F,heat}$ building heat emission at local scale estimated from the heating-degree-day model (Järvi et al., 2011), $fr_{nonheat}$ fraction of fossil fuels used for building energy other than heating (e.g. the use of gas stove for cooking), $Q_{F,base}$ non-temperature related anthropogenic heat flux (W m$^{-2}$) including heat emissions from traffic, human metabolism and electricity usage, $fr_{QF,base,BEU,d}$ the fraction of the $Q_{F,base}$ coming from building energy use on weekdays or weekends, and $E_{CO2perJ}$ the EF for fuels in building heating and energy use (μmol $CO_2$ J$^{-1}$)."

*Eq. 15: Just to be clear, the model does not take into account LAI variability in Fres estimation, right?*

No, the $F_{res}$ estimation does not consider LAI variability.

*Line 156: I suggest you state here that the main model domain is the 1km radius circle around the tower.*

Modified text (Lines 156−158):

"The model domain is a 1 km circle around the 325 m meteorological tower constructed by Institute of Atmospheric Physics, Chinese Academy of Sciences (IAP tower, 39◦58' N, 116◦22' E, 60 m above sea level) located in the 6th Ring area of Beijing, China (Fig. 1 a)."

*Lines 170 - 171: A large fraction of the wind directions are omitted from the analysis. This could affect significantly the land cover fractions "seen" by the observations. Moreover, the wind seasonal and diurnal wind direction patterns are not presented in any way. This information is very crucial when interpreting the measured Fc. The LC fractions of the SUEWS domains can be very different to the actual flux source area defined by the wind patterns and the omitted wind sectors. It would be useful to add the omitted wind directions as shaded areas in Fig. 1c and also include a wind rose in Fig. 1 to give an overview of which directions are affecting more the observations (supplementary to the detailed analyses suggested in major concern No. 2).*

We compared land cover fractions before and after wind direction filtering (WDF), and found that the land fractions do differ (Table S1, Fig. R2). To examine the impact of land fractions chosen on simulated turbulent fluxes, we conducted additional model cases over the "actual surface" without the filtered wind directions (the remaining sectors after WSF). We found that SUEWS output is very similar to the output with the original SUEWS model domain (Fig. S1). Furthermore, the source area is elliptic and not linear with wind direction, and thus removing narrow wind direction bands removes areas which are more or less "seen" by observations. Thus, we want to include these narrow bands to our model simulations. Therefore, we kept the land cover fractions of the "original surface" in the main text.

Table S1 and Fig. S1 were added to the Supplementary materials. Meanwhile, we edited Fig. 1 and added two subplots, providing the information regarding (1) the wind sectors that have been omitted; (2) the wind direction frequency by season; (3) the road information within the study area.

Added text (Lines 170−171):

"… (3) Wind direction filtering: the wind directions with building heights over 50 m (112-128, 160-243, 314-3°) are removed (Kokkonen et al., 2019) **(Fig. 1 b, c).**"

Added text (Line 176):

"Wind directions are mainly from the S-NW sectors and the NE-E sectors before the implementation of the wind direction filtering (Fig. 1 e). In winter, wind from the west is more frequently seen than from the east compared to the other seasons. Due to the wind direction filtering, the actual flux source area 'seen' by the EC measurement is biased from the 1 km radius circle around the IAP tower. The vegetation fraction is 31% in the remaining sectors combined, as compared to 29% in the entire 1 km radius circle (Fig. 1 b–c, Table S1). The model performance in turbulent flux modelling with the land fractions for the remaining sectors is similar to the entire circle (Fig. S1). Therefore, only the model results using the land fractions of the entire circle are demonstrated in the main text."

[Figure]

Figure R2. Land cover of the study area. The red line denotes the 1 km radius circle around the IAP tower. The white solid lines denote the wind direction sectors where the turbulent fluxes are filtered out for quality control as described at Section 3. The area that is not covered by geometries are considered as paved surfaces.

*Lines 177 – 181: In addition to the criticism on this method (major concern No. 1), this paragraph is not very clear for the reader.*

The paragraph has been rephrased (please see the response to the major concern No. 1).

*Lines 195 – 197: Is here the right place to describe the method for heat storage flux?*

We started a new sub-section to describe the storage heat flux.

Added text (starting from Line 195):

**"4.3 Storage heat flux"**

"To calculate the storage heat flux, Objective Hysteresis Model (OHM) is used (Grimmond and Oke, 1999). The coefficients for all the surface types follow the previous study by Kokkonen et al., 2019, except for the paved surface. A large portion of the paved surface is asphalt in the study area. Thus, the coefficients are set to the weighted average values of asphalt surface (AN99) following Ward et al. (2016)."

Modified text (Lines 199−200):

"In this study, local parameters of traffic, population dynamics, and building energy use are incorporated in order to estimate anthropogenic heat flux and $CO_2$ emissions."

*Lines 202 – 203: I guess it is due to my lack of understanding of the units, but could you give more clear explanations on how these traffic rates are calculated?*

Modified text (Lines 201−203):

"The annual mean weekday and weekend diurnal cycle of traffic rate for each road link in 2017 in the study domain are extracted from a dataset based on an extensive road traffic monitoring network (Yang et al., 2019). For weekends and weekdays, the diurnal traffic cycles are calculated independently. The total hourly traffic rate (veh km hour$^{-1}$) is calculated as the sum of the traffic rates, i.e., the product of traffic volume (veh hour$^{-1}$) and the road link length (km) from all the road links in the study area. The hourly traffic rates are then summed up to the total daily traffic rates (veh km day$^{-1}$), and divided by the total modelling area, yielding $Tr_d$. Finally, the diurnal traffic profiles ($H_{T,d}$) are obtained by normalizing the diurnal cycle of the total hourly traffic rate (Table 1, Fig. 2).

*Lines 216 – 218: This is an important part to interpret the measured Fc. You state that the emissions from the boiler plants are "very likely" to be observed by the EC, but you do not give any more information of why do you assume that. You also say that you have located at least 3 boiler plants in the surroundings. Could you give more information on how have you located them, where they are (add them in Fig. 1) and what is the height of their chimneys? This information can help to assess if these emissions are actually affecting the observations. Moreover, a wind sector analysis of the observed Fc (as suggested in major concern No. 2) can provide evidence on the effect of point source emissions on the data.*

We thank the reviewer for the important insight, and we agree that the information about the nearby boiler plants should be added.

[Figure]

Figure R3. Boiler plants for district heating near the IAP tower and their chimney height (m). The red solid line denotes the 1 km radius circle around the IAP tower. Note that the locations of the boiler plants might be inexhaustive.

We investigated the boiler plants through interviewing the residents, residential district property offices, and heat supply companies in the study area. Although the locations of some boiler plants are known through interviews (Fig. R3), those boiler plants located in the districts and schools inaccessible to outsiders are missing.

Considering the information about boiler plants might be incomplete, they are not marked to the picture in main text. Nonetheless, we found that boiler plants are very common in the nearby neighborhoods around the IAP tower. For the known 11 boiler plants, 8 of them have a chimney height lower than 20 m, indicating that the observational $F_C$ (measured at 47 m) very likely contained the contribution of district-level heating during winter (Fig. R3).

Unfortunately, the heating capacity and detailed information regarding fuel combustion for each boiler plant are unknown or restricted from access. Therefore, it is challenging to treat boiler plant $CO_2$ emission as point sources.

As an alternative, the $CO_2$ emissions contributed by district-level heating are calculated from anthropogenic heat estimate, and specifically, from the heating demand related to air temperature, district-level heating fraction, the share of fuel type, emission factors collected from a yearbook and other studies (see also Lines 211−227).

Modified text (Lines 214−215):

"Boiler plants are very common: over 5000 coal-fired and 1000 gas-fired heating boilers are located surrounding the populated areas in 2014 (Cui et al., 2019)"

Modified text (Lines 216−219):

"We investigated the boiler plants near the IAP tower through interviews, and found that there were at least 11 of them located at multiple directions within 1.5 km distance from IAP tower. For the known boiler plants, 8 of them have a chimney height lower than 20 m. Thus, their $CO_2$ emissions are very likely to be observed by EC at 47 m during heating season. Unfortunately, the heating capacity and detailed information regarding fuel combustion for each boiler plant are unknown or restricted from access. Therefore, it is challenging to treat the boiler plant $CO_2$ emission as point sources. As an alternative, SUEWS first estimates the anthropogenic heat release from heating $Q_{F,heat}$ and then converts the heat into local $CO_2$ release using the EF and the fraction of fossil fuels used for heating $fr_{heat}$ (Eq. 13)."

*Line 231: The 20 % used as frnonheat sounds an arbitrary choice. Could you explain in more detail how you end up with this number?*

We estimated $fr_{nonheat}$ from an observational study over household indoor $CO_2$ release in Beijing. "Household fuel combustion, mainly from cooking, takes place throughout the year, but its $CO_2$ emission are relatively small. An observational study shows that $CO_2$ emitted from fuel combustion in household cooking contributes only 6% of the indoor $CO_2$, and this percentage is low, compared with contribution by human metabolism (30%) (Shen et al., 2020)" (Lines 228−231). In other words, the ratio of

$CO_2$ emitted by non-heating household fuel combustion to metabolic $CO_2$ release was 1:5. We have examined several SUEWS model runs with different values of $fr_{nonheat}$ and made sure the ratio of output building $CO_2$ emissions to metabolic $CO_2$ release during non-heating season to remain the ratio of 1:5, and $fr_{nonheat} = 20\%$ was therefore determined.

There is also a direct approach to estimating $fr_{nonheat}$. Beijing Statistical Yearbook showed that urban household living consumed liquefied petroleum gas $27.9 * 10^7$ kg of coal equivalent (kgce), gas $17.1 * 10^8$ kgce, and electricity $20.46 * 10^8$ kgce, indicating that ~50% of the household energy use involves on-site $CO_2$ emissions, i.e., $fr_{nonheat} = 50\%$ (BMBS, 2017).

The first approach estimates the $fr_{nonheat}$ from independent indoor $CO_2$ emission observations. The second approach is closer to the original definition of $fr_{nonheat}$. These two approaches result in a difference in the annual $F_C$ estimate of 0.35 kg C m$^{-2}$ yr$^{-1}$ in all the model cases. After a discussion, we finally decided to update the $fr_{nonheat}$ to 50% to all the cases. All the related figures and statistics have been updated and presented in the revised manuscript.

Modified text (Lines 229−231):

"Statistics showed that urban household living consumed liquefied petroleum gas 27.9 $\times 10^7$ kg of coal equivalent (kgce), gas $17.1 \times 10^8$ kgce, and electricity $20.46 \times 10^8$ in 2016 (BMBS, 2017), indicating that 50% of the household energy use involves on-site $CO_2$ emissions. Therefore, the non-heating fraction ($fr_{nonheat}$) is set to 0.5"

*Table 1: Should the unit for traffic EFs be kg km-1 veh-1?*

Thank you for noticing the mistake. The unit has been corrected.

*Lines 245 – 252: I have the impression that the description of the case base can be much simpler by referring to Table 2 parameters.*

We understand that this paragraph appears to be a little too lengthy. However, we believe the details about the control run (case **base**) should be provided to readers completely. Although we did adopt the almost all the parameters used by Kokkonen et al. (2019), there are still some exceptions that deserve explanation. For example, Kokkonen et al. did not have the parameter of maximum photosynthetic rate ($F_{pho,max,grass}$) for grass/lawn (2019), but we have derived this parameter from observation as shown in the Appendix.

*Line 255: Table 4 is referred before Table 3.*

Thank you for finding this mistake. The table captions have been corrected.

*Table 4: I am not familiar with the LAI model of Eq. 1, but it seems that all the parameters have some physiological meaning. The optimised parameters presented in Table 4 are very different to the original ones (the signs of both ω1 are even inverse) and I am wondering if they still make sense in terms of plant physiology.*

It is true that the terms $T_{base,GDD}$, $T_{base,SDD}$, $GDD_{full}$, and $SDD_{full}$ have some physiological meaning related to the growing-degree-day (see Lines 77−80).

However, the terms $\omega_{1,GDD}$, $\omega_{2,GDD}$, $\omega_{1,SDD}$ and $\omega_{2,GDD}$ do not have physiological meanings, and they are curve factors that determine the shape of LAI time series. Moreover, their values are basically within the range reported by Omidvar et al. (2022). We reported them as how they were when obtained from the optimization approach.

These parameters have been updated using the Landsat 7 LAI (see also the response the general concern No.1).

*Line 290: add comma after "LAI behaviour"*

Thank you for the language suggestion. We have revised the original paragraph under one of the suggestions from another reviewer, and therefore the phrase "LAI behaviour" has been deleted.

*Section 5.1, Figure 3: The seasonal LAI patterns of MODIS and the optimised model are a bit strange. I find it hard to believe that the deciduous species and the grasses present so slow greening phase in spring and summer that they reach full greenness in late July. The original SUEWS patterns, even though they present very steep changes, are more realistic. Could you provide any ground truth to support the optimised model? See also main concern No. 1.*

Currently, we agree that a higher-resolution LAI product should be used to better represent the actual vegetation seasonal dynamics in the study. Therefore, we chose the Landsat 7 LAI as the input data to the CMA-ES optimization (see also the response to the general concern No. 1).

*Section 5.2: I wonder if this section is relevant in this study since the vegetation parameterization seems to not have any effect on the model results.*

This section has been moved to Appendix (see also the response to general concern No. 3).

*Lines 313 – 314: I am confused by this statement, if the built surface emissivity is underestimated then the LWup would also be underestimated, but the opposite is true in your case. Since the overestimation of LWup occurs mostly during day, it might be that it is indirectly introduced in the model by the Kdown overestimation (Offerle et al., 2003).*

Thank you very much for noticing the mistake.

Modified text (Lines 314−315):

"The values of emissivity of building materials used in SUEWS might be slightly **higher** than in reality."

*Line 322: The performance of the case LAI seems very similar to the case base in QE estimation.*

With the previous optimized LAI parameters for case **LAI**, the $R^2$ increased and RMSE decreased slightly when compared to case **base**. Currently, with the new LAI parameters, case **LAI** is very similar to case **base**. The statement has been rephrased.

Modified text (Line 322):

"With the optimized LAI (case **LAI**), model performance in $Q_E$ remains virtually unchanged, with RMSE (12.1–94.1 W m$^{-2}$) and $R^2$ (0.17–0.53) when compared to case **base** (with RMSE 11.7–96.1 W m$^{-2}$ and $R^2$ 0.20–0.51) (Fig. 7 a–c)."

*Lines 327 – 333: It could also be that that the opt LAI is underestimated during spring and autumn.*

With the new LAI parameters, the opt LAI during spring and autumn increased. The case **gs_LAI** overestimates $Q_E$ in spring and summer (see also the response to the general concern No.1). However, it still underestimates $Q_E$ in winter. We believe this might be attributed to the underestimation of anthropogenic water vapor in winter.

*Lines 334 – 339: It would worthwhile to include some small discussion on the effects of the other energy balance parameters in QH performance. QF and ΔQs could as well affect the results of QH.*

We agree.

Added text (Line 339):

"$Q_H$ is also influenced by $Q_F$. Nighttime $Q_F$ in summer might be overestimated, leading to the overestimation in $Q_H$. Turbulent heat fluxes are also related to $\Delta Q_s$. Both $Q_E$ and $Q_H$ correlate with $\Delta Q_s$ negatively (Eq. 2, Eq. 9). For instance, $Q_E$ is underestimated while $Q_H$ overestimated at noon in summer in case **gs_LAI**. The decrease in $\Delta Q_s$ will lead to a simultaneous increase in $Q_E$ and $Q_H$, lowering $Q_H$'s

bias while increasing $Q_E$'s. Therefore, the adjustment of $\Delta Q_S$ can hardly improve the $Q_E$ and $Q_H$ modelling at the same time."

*Line 336: Delete "(SON)".*

The word "(SON)" has been deleted.

*Section 5.4.2: The title "model uncertainties" is misleading. This section does not quantify or report model uncertainties but instead it discusses the results of this study compared to previous literature. I suggest to merge this section with the previous one.*

This section has been merged to the previous one.

*Line 377: Plural: "Building emissions are ...".*

This phrase has been corrected accordingly.

*Lines 400 – 415: This paragraph has a lot of repeating and unclear phrasing, it can be rewritten in a more concise and clear way.*

Modified text (Lines 400−415):

"The turbulent flux modelling is usually evaluated over a fixed extent such as a circle with a certain radius to approximate the flux source area (Demuzere et al., 2017; Järvi et al., 2019). However, when a circle with the radius ≥1000 m is selected to approximate the ≥80% footprint fetch in our study, SUEWS does not give the closest estimate of annual $F_C$. … First, the accumulated footprint area of the observed fluxes is irregular in shape and vary with time (Liu et al., 2012). Second, the relative contribution to flux from the land surface decreases as the distance to the measurement instrument increases (Christen et al., 2011; Rebmann et al., 2005). Thus, when the modelling domain is a 1000 m radius circle, the model might underestimate the relative contribution from the adjacent vegetated surface, and overestimate the contribution of the traffic hot spot at the edge of 80% footprint fetch."

*Lines 409 – 410: What do you mean when stating that it is challenging to change the model domain when the soil processes are involved?*

The soil processes include several hydrological processes, such as runoff and drainage, they are connected to the turbulent flux modelling, such as $Q_E$ and photosynthetic $CO_2$ uptake by vegetation. They are continuous processes, which cannot be restarted halfway. Therefore, it is challenging to allow the model domain to accommodate flux source area changes in time. We decided to delete this sentence to avoid confusion.

*Lines 422 – 424: Make clear here that you talk about the local emissions within the study area, otherwise these values would not make any sense.*

We agree.

Modified text (Line 422):

"The contribution of the local building emissions within the study area is more variable among cities: …"

*Line 432: Is this true because the radiation model does not take into account such changes?*

No, the radiation model does not consider $g_{max}$ or LAI dynamics. We decided to remove this sentence from the conclusions.

Deleted text (Line 432):

"Radiation flux modelling performs well without fine-tuning and it is hardly influenced by $g_{max}$ and LAI."

*Line 433: "… great sensitivity to gmax and the behaviour of LAI". LAI seems not to be so important.*

Modified text (Line 433):

"The model performance of heat fluxes ($Q_E$ and $Q_H$) is more sensitive to the adjustment of $g_{max}$ than to the change of LAI seasonal dynamics in our study area."

*Line 440: Replace "In comparison of" with "Compared to".*

The phrase has been rephrased accordingly.

*Lines 445 – 446: This sentence can be omitted from the conclusions.*

The sentence has been removed accordingly.

*Lines 453 - 454: Rephrase this sentence to be clearer.*

Modified text (Lines 453−454):

"We believe that the bottom-up approach to model $F_C$ by SUEWS can be a promising tool in capturing the $CO_2$ emission hot spots, quantifying the relative contribution of the local $CO_2$ sources, and assisting to mitigate urban $CO_2$ emissions."

*Line 468: The year is missing from Hansen et al.*

The missing information has been added.

*Line 473 and later: LAI at "tree level" is hardly defined as a concept. LAI is always relative to the ground surface area. When you define seasonal LAImax, LAImin for a vegetation species or type you usually assume a homogeneous area totally covered by this species/type.*

We agree. The phrase "tree level" was replaced by "canopy level" since the measured LAI usually represents the canopy within a certain extent.

Modified text (Lines 473−474):

"We note that original LAI might be noticeably lower than the measured LAI at the canopy level over a homogeneous vegetated surface."

*Line 485: Singular: "The observed LAI **is** linearly …".*

The phrase has been corrected accordingly.

*Lines 527 – 528: Do you mean that you used the night-time flux partitioning method by Reichstein et al. (2005)? Be more specific.*

The extrapolation of night-time temperature dependency of ecosystem respiration is used to partition the $CO_2$ flux measured. However, the exact model we used is different from the model mentioned by Reichstein et al., (2005). Instead, the simple exponential fitting was applied.

To make it clear, we modified lines 527 – 528:

"The biogenic $CO_2$ flux ($F_{C,bio}$) from EC measurements was partitioned into $F_{res}$ and $F_{pho}$ using the night-time temperature dependency of $F_{res}$. The nighttime $F_{C,bio}$ was considered as nighttime $F_{res}$, and it was related to the observed air temperature by fitting the exponential model,

$$F_{res} = a_{grass} \cdot exp(T_{air} b_{grass}).$$

The nighttime temperature dependency of $F_{res}$ was then extrapolated to daytime, and $F_{pho}$ was then calculated by subtracting $F_{res}$ from $F_{C,bio}$,

$$F_{pho} = F_{C,bio} - F_{res}.$$"

*Line 536: Add "applying a" before "bootstrapping".*

The phrase has been added accordingly.

*Line 542: Plural: "the species in the modelled area **are** known…"*

Thank you for noticing the grammar mistake. The phrase has been corrected.

**References**

Zheng Y, Liu H, Cheng X. Datasets for simulating heat and CO2 fluxes in Beijing using SUEWS V2020b [DS]. Zenodo, 2022. https://doi.org/10.5281/zenodo.7427360.

Chen W, Jia X, Zha T, et al. Soil respiration in a mixed urban forest in China in relation to soil temperature and water content[J]. European Journal of Soil Biology, 2013, 54: 63-68.

Zhang X, Zhang X, Li G. The effect of texture and irrigation on the soil moisture vertical-temporal variability in an urban artificial landscape: A case study of Olympic Forest Park in Beijing[J]. Frontiers of Environmental Science & Engineering, 2015, 9: 269-278.

Masek J G, Vermote E F, Saleous N E, et al. A Landsat surface reflectance dataset for North America, 1990-2000[J]. IEEE Geoscience and Remote sensing letters, 2006, 3(1): 68-72.

Boegh E, Soegaard H, Broge N, et al. Airborne multispectral data for quantifying leaf area index, nitrogen concentration, and photosynthetic efficiency in agriculture[J]. Remote sensing of Environment, 2002, 81(2-3): 179-193.

Huete A R, Liu H Q, Batchily K V, et al. A comparison of vegetation indices over a global set of TM images for EOS-MODIS[J]. Remote sensing of environment, 1997, 59(3): 440-451.

Crawford B, Christen A. Spatial source attribution of measured urban eddy covariance CO2 fluxes[J]. Theoretical and Applied Climatology, 2015, 119: 733-755.

Beijing Municipal Bureau of Statistic, Beijing statistical yearbook[Z]. China Statistics Press, 2017.

Omidvar H, Sun T, Grimmond S, et al. Surface Urban Energy and Water Balance Scheme (v2020a) in vegetated areas: parameter derivation and performance evaluation using FLUXNET2015 dataset[J]. Geoscientific Model Development, 2022, 15(7): 3041-3078.

**Date: 08 Jul 2023, Iteration: Correction**

Dear Editor and Reviewers,

We appreciate the approval from the reviewers, and we greatly thank the Editor and Reviewers for taking the time and effort on our submitted manuscript. According to the comments from the Anonymous Reviewer #2, the manuscript has been revised carefully. The Reviewer's comments are in italics and our response is plain text. Our responses are shown below.

**Response to Anonymous Referee #2 (July 8 2023)**

*I would like to thank the Authors for taking all my concerns and suggestions into account, performing major revisions to the manuscript and providing detailed answers to my questions and comments. All my major concerns and specific comments have been adequately addressed and I believe that the revised manuscript is considerably improved. I have only some minor comments on the revised manuscript:*

We sincerely appreciate all the comments from the Reviewer, which helped us to improve the quality of the manuscript further. Our responses are as follows:

*Eq. 11, Eq. 12, Eq. 13: I am a bit confused with the subscripts h and d. I understand that h denotes hour and d the type of day (weekdays or weekend). If so, the daily average population density $p_{h,d}$ should not have the subscripts d and h, right? Also, in Eq. 12 and 13 the subscripts are missing from some parameters, but maybe it is for simplicity.*

Thank you for noticing the mistake. In Eq. 11, $p_{h,d}$ should be $p_d$. The equation and the explanation have been revised accordingly.

In Eq. 12 and 13, for those parameters that are independent of hour or type of day (weekdays or weekend), the subscripts h or d are omitted accordingly. For example, the parameter $Q_{F,base}$ in Eq. 13 is independent of both hour and type of day, so it does not have the subscripts h or d. Additionally, the hourly variable output by SUEWS (e.g., $Q_{F,heat}$ in Eq. 13) does not have the subscripts for simplicity.

*Lines 187 – 191: Maybe this part would be more relevant to be moved to Section 4.2.*

This part was moved to Section 4.2 accordingly.

*Line 261: Please add the units kgce after the electricity consumption value.*

The unit was added accordingly.

*Line 382: Adding a reference to Fig. 6 at the end of this sentence would make the statement clearer.*

A reference to Fig. 6 was added at the end of this sentence accordingly.

*Line 386: "On the other hand, ..."*

The phrase has been rephrased accordingly.

*Lines 393 – 425: The discussion on Fc performance is now improved. However, I believe that there is much more information hidden behind the tower Fc measurements that is not yet fully discussed. Figure S2 is very insightful regarding the Fc behavior around the site. It is very distinctive that the afternoon Fc peak, which is obvious all year round, comes from the NE direction, while the morning peak during cold months is coming from both NE and NW. Photosynthesis signal during summer is coming from SW-W directions but these wind directions are not so frequent during this period, so probably photosynthesis is not very obvious on the observations. The distinctive afternoon peak from NE could be either traffic from this secondary main road or a building source that is active all year round. Probably the first, since the road is very close to the tower site from this direction and its effect on the measurements would be strong. If this is the case, then the traffic profile used in SUEWS is not so representative for the study area. Maybe there is more traffic or/and more congestion on this road during afternoon compared to morning. The building heating emissions seem to originate mainly from NW and SE directions. The diurnal pattern is consistent with the one used in SUEWS, but is seems that building emissions are also evident around midnight.*

*I would recommend the Authors to include such discussion in the manuscript. It is expected that each urban site is very unique and it is difficult for model parametrization. Local insights are always useful for future research.*

We thank the Reviewer for the constructive insight in the $F_C$ measurements. We added the analysis of $F_C$-wind direction variation to the main text. Meanwhile, we also state that SUEWS cannot consider this $F_C$'s wind-direction dependency as it simulates the overall flux from the simulation domain.

Added text (Lines 393–402):

"… First, the underlying seasonal variation in the diurnal wind pattern makes the $F_C$ diurnal cycle from the NW quadrant more "seen" in winter and spring, while the diurnal cycle from SE more "seen" in summer and autumn (Fig. S2 a–d). Second, the observed $F_C$ varies noticeably with wind direction (Fig. S2 e–h). An evident afternoon and evening $F_C$ peak is observed from NE and SE throughout the year. This might be attributed to the relatively high traffic volumes and more severe traffic congestion in the afternoon and evening than in the morning, especially at the nearby crossroads (Fig. 1 d). The morning $F_C$ peak comes from only NE and NW, and only during winter months, suggesting that there might be seasonal variation in the traffic pattern that is not captured by SUEWS. The noticeable increase in $F_C$ in winter from the SE and NW indicates that the building heating emissions might originate mainly from these two sectors. …"

*Fig 7: Maybe consider to use the abbreviations of Eq. 10 for each Fc component on this figure for consistency.*

The figure and its caption have been updated accordingly.

*Line 466: This sentence is not very clear. Can be rephrased as: "gmax was parametrized according to leaf-level observations of vegetation species in Beijing".*

The sentence has been rephrased accordingly.

*Line 474: "This shows that the model…".*

The word "that" has been added.

*Line 475: "followed by human metabolism (14 -18 %), buildings (11-14%) …".*

The word "human" has been added, and the word "building" has been replaced with "buildings".

*Line 478: "We highlight the importance of choosing …".*

The word "in" has been replaced with "of".

*Lines 502 – 505, Fig. A1: Noisy Landsat pixels are probably still present despite the spike removal step. The abrupt LAI drop and recovery during summer is not realistic. It is probably due to the effects of clouds or bad illumination conditions. I am not sure if cloud masking or other quality flags are provided with this product, but I would suggest the Authors to check.*

*In any case, I would not suggest that the method should be revisited, since the optimization method provides a LAI seasonal pattern that ignores the bad Landsat values. Maybe there is some underestimation of the predicted values, but since the LAI pattern is subsequently normalized between minimum and maximum LAI in the model, this would not be a problem. However, I strongly suggest that the Authors comment on these problematic issues (very common in satellite data) and argue why the problematic Landsat values are kept in the analysis.*

We thank the Reviewer for the suggestion on the quality check of the Landsat data. The noise might originate from the cloud or shadow. To our knowledge, the Landsat 7 pre-processing does not include cloud masking as the cloud masking techniques on bright surfaces (e.g., bare sand, impermeable surfaces) perform poorly. Furthermore, there are no cloud mask functions as an option during the process of the LAI data extraction. Considering that the noise has little impact on the CMA-ES optimization, we believe that the noise would not be a problem.

Modified text (Lines 518–520):

"Despite the spike removal step, noise is observed in the LAI time series such as the abrupt drop and recovery during summer. Smoothing techniques are used to extract valid information from the satellite data with noise. In other words, the overall pattern (e.g., the moving average) of the time series is the information of interest, and CMA-ES has successfully reproduced the overall pattern out of the input LAI time series contaminated by noise (Fig. A1 a). Moreover, the LAI has been "scaled" as shown in step 3 to allow the output LAI to reach 5–6 $m^2$ $m^{-2}$ in order to avoid the underestimation of the predicted values. Therefore, the noise in the input LAI during summer is kept since it has little impact on the outcome."